

# The Canadian atmospheric transport model for simulating greenhouse gas evolution on regional scales: GEM-MACH-GHG v.137-reg

Jinwoong Kim, Saroja M. Polavarapu, Douglas Chan, and Michael Neish

Climate Research Division, Environment and Climate Change Canada, Toronto, Ontario, M3H 5T4, Canada

*Correspondence to*: Jinwoong Kim (Jinwoong.kim@canada.ca)

**Abstract.** In this study, we present the development of a regional atmospheric transport model for greenhouse gas (GHG) simulation based on an operational weather forecast model and a chemical transport model at Environment and Climate Change Canada (ECCC), with the goal of improving our understanding of the high spatio-temporal resolution interaction between the

atmosphere and surface GHG fluxes over Canada and the United States. The regional model uses 10 km x 10 km horizontal grid spacing and 80 vertical levels spanning the ground to 0.1 hPa. The lateral boundary conditions of meteorology and tracers are provided by the global transport model used for GHG simulation at ECCC. The performance of the regional model and added benefit of the regional model over our lower resolution global models is investigated in terms of modelled $CO_2$ concentration and meteorological forecast quality for multiple seasons in 2015. We find that our regional model has the

capability to simulate high spatial (horizontal and vertical) and temporal scales of atmospheric $CO_2$ concentrations, based on comparisons to surface and aircraft observations. In addition, reduced bias and standard deviation of forecast error in boreal summer are obtained by the regional model. Better representation of model topography in the regional model reduces transport and representation errors significantly compared to the global model, especially in regions of complex topography, as revealed by the more precise and detailed structure of the $CO_2$ diurnal cycle produced at observation sites and in model space. The new

regional model will form the basis of a flux inversion system that estimates regional scale fluxes of GHGs over Canada.

## 1 Introduction

The global mean atmospheric carbon dioxide ($CO_2$) concentration or mixing ratio (in mole fractions of dry air) has been increasing since the industrial revolution mainly due to anthropogenic emissions into the atmosphere, while terrestrial and oceanic uptake moderate the increase of $CO_2$ in the atmosphere (Canadell et al., 2007; Le Quéré et al., 2009). Apart from this

global increase, information about each component affecting the global carbon budget and its uncertainties are estimated and updated regularly at the global scale, using a wide range of methods and data (Le Quéré et al., 2009; 2018). Since the ocean $CO_2$ sink have been increasing constantly in line with the increased $CO_2$ in the atmosphere (Le Quéré et al., 2018), the interannual variability of the $CO_2$ growth in the atmosphere is primarily attributed to that of terrestrial fluxes. Recently, the mean annual atmospheric $CO_2$ growth rate reached a record high mainly due to the impact of El Niño Southern Oscillation on





the interannual variability of biospheric fluxes (Buchwitz et al., 2018) and increased net biospheric respiration in the tropics (Liu et al., 2017). Globally, increased $CO_2$ and temperature are positively or negatively associated with terrestrial uptake by enhancing photosynthesis or respiration (Fernández-Martínez et al., 2018). Regionally, however, the carbon balance of Canadian boreal forest and its impact on global carbon budget is highly uncertain and the ecosystems of Canada are vulnerable

to a changing climate (Kurz et al., 2013; Bush and Lemmen, 2019). Therefore, correctly accounting for biospheric fluxes over Canada is important for understanding both the global and regional carbon cycles.

Surface sources and sinks of $CO_2$ can be estimated through inverse modelling using atmospheric $CO_2$ concentrations as a constraint to adjust prior fluxes so as to minimize the difference between the modelled $CO_2$ concentrations and observed values (Ciais et al., 2010). Many atmospheric inversion studies have been conducted to quantify surface $CO_2$ fluxes on both

global and regional scales (Tans et al., 1990; Gurney et al., 2002; Peters et al., 2007; Lauvaux et al., 2008; Lauvaux et al., 2012b). Although there is a consensus of estimated fluxes at the global scale, significant discrepancies among different inversion system results still exist, especially in partitioning terrestrial fluxes at continental scales (Peylin et al., 2013) due to the contribution of atmospheric transport model errors and prescribed fossil fuel emissions (Peylin et al., 2011; Gaubert et al., 2019).

In an atmospheric inversion of $CO_2$, the transport model plays a key role in transforming the surface $CO_2$ flux information into atmospheric $CO_2$ concentrations and can be used as a verification tool for estimated surface $CO_2$ fluxes (Ciais et al., 2010; Nisbet and Weiss, 2010; Bergamaschi et al., 2018). The errors caused by an imperfect transport model can introduce biases and uncertainties into estimated fluxes during the inversion process (Law et al., 1996; Gloor et al., 1999; Engelen et al., 2002; Houweling et al., 2010; Chevallier et al., 2010, 2014; Locatelli et al., 2013). Such errors may arise from

a variety of sources: model formulation, meteorological fields and representativeness errors. Model formulation errors may arise from processes associated with parametrizations of vertical mixing within the planetary boundary layer (PBL) (Lauvaux and Davis, 2014), vertical mixing between the PBL and the free troposphere (Stephen et al., 2007), isentropic transport (Parazoo et al., 2012; Barnes et al., 2016), synoptic scale variations due to advection and convection (Parazoo et al., 2008) and mid-latitude storm tracks (Parazoo et al., 2011). In fact, the impact of synoptic and mesoscale transport on the variability of

$CO_2$ is comparable with that of surface fluxes (Chan et al., 2004). Since an atmospheric transport model is driven by meteorology, uncertainties in meteorological model and observations is another important source of error in the transport of tracers (Liu et al., 2011; Miller et al., 2015; Polavarapu et al., 2016). Finally, representation error is also a source of errors in inversions. The mismatch between coarse resolution transport model simulations and observations from real $CO_2$ field impacts the ability to resolve sub-grid scale variability of $CO_2$. In particular, unresolved synoptic and mesoscale processes increase

representation error (Engelen et al., 2002).

The sparseness of the $CO_2$ observation network used in inversions is another major contributing factor to the uncertainty in estimated fluxes. Increasing the density of the surface observation network is beneficial for reducing uncertainty and improving the accuracy of retrieved fluxes in the context of both global (Bruhwiler et al., 2011) and regional inverse modelling (Lauvaux et al., 2012a; Schuh et al., 2013). Since a number of new measurement sites have been established over





Canada and the US in recent decades (e.g. Worthy et al., 2005; Andrews et al., 2014; Bush et al., 2019), it should now be possible to obtain optimized fluxes on finer spatial scales and with reduced uncertainties. However, in order to better interpret information from spatially dense observation networks which contain information on strongly varying biospheric fluxes and strong sources of anthropogenic emissions, a high resolution atmospheric transport model capable of capturing these signals
is needed.

Resolving the fine scale spatial and temporal variability of $CO_2$ generated by heterogeneous land surface and complex topography, which is not resolved by typical grid sizes of global models, is the primary motivation for regional scale inverse modelling. Increased horizontal resolution could alleviate transport and representation errors and thus improve simulations of synoptic variations of $CO_2$ concentrations (Patra et al., 2008; Remaud et al., 2018). Indeed, Gerbig et al. (2003) suggested that
in order to resolve spatial variations of $CO_2$ in the PBL over a continent, a horizontal grid spacing no larger than 30 km is required. In addition, Pillai et al. (2011) showed that a maximum horizontal resolution of 12 km is required to represent the variability of $CO_2$ concentrations especially over mountainous or complex terrain. To this end, several studies focusing on forward $CO_2$ simulation at regional scales were carried out, using different models and configurations over various regions of interest. One approach to simulate atmospheric $CO_2$ concentration at finer spatial and temporal resolution is using zooming or
nested domains within a global model (Krol et al., 2005; Lin et al., 2018). Another option is to use a regional atmospheric transport model. Various kinds of regional scale modelling studies have been conducted for the mid continental region of North America (Díaz-Isaac et al., 2014), south west France (Ahmadodv et al., 2007; 2009), western Europe (Kretschmer et al., 2014) and East Asia (Ballav et al., 2012). By increasing horizontal and vertical resolutions, regional models have an advantage over global models in terms of simulating $CO_2$ concentrations as shown by intercomparison experiments (Geels et al., 2007;
Pillai et al., 2010; Díaz-Isaac et al., 2014).

At Environment and Climate Change Canada (ECCC), a carbon assimilation system (EC-CAS) is under development in order to estimate surface greenhouse gas (GHG) states and fluxes. To this end, a GHG forward modelling system which includes coupled meteorology and tracer transport model with full model physics, namely GEM-MACH-GHG (Polavarapu et al., 2016), has been developed. GEM-MACH-GHG is based on an operational weather forecast model, Global Environmental
Multiscale model (GEM) at Canadian Meteorological Centre (CMC) (Côté et al. 1998a, b; Girard et al., 2014), and a chemical transport model with complete tropospheric chemistry, GEM-Modelling Air quality and Chemistry (GEM-MACH) model (Moran et al., 2010; Robichaud and Ménard, 2014; Makar et al., 2015), although the tropospheric chemistry module is not used in GEM-MACH-GHG simulations. GEM-MACH-GHG with 0.9° horizontal grid spacing is capable of simulating $CO_2$ concentrations over the globe acceptably well in comparison with in-situ and surface-based column averaged $CO_2$
observations. GEM-MACH-GHG was also used to investigate the uncertainty of $CO_2$ transport across different global transport models (Polavarapu et al., 2018), and was tested with the Canadian Land Surface Scheme and Canadian Terrestrial Ecosystem Model (CLASS-CTEM) in order to be able eventually to consistently simulate atmosphere-land exchange of $CO_2$ over the globe (Badawy et al., 2018). While a limited area version of the GEM model exists for operational weather and air quality



forecasting, the ability to simulate GHGs on a regional model domain over a continental region had not been developed before now.

In this paper, in order to obtain a better understanding of the variability of atmospheric $CO_2$ concentration at finer spatio-temporal scales over a continental region during a relatively long time period, a regional scale atmospheric transport

model for GHG simulation based on GEM-MACH-GHG is developed and tested. As a first step, $CO_2$ simulations with a 10 km grid spacing are performed for the year 2015, on a domain covering most of Canada and the US. The performance of the new model is investigated using meteorological and $CO_2$ concentration observations. In addition, the added benefit of the regional model over the global model in terms of $CO_2$ simulation as well as weather forecasts is investigated. The article is organized as follows. A description of the model, data and methodologies used in this study is provided in Sect. 2. In Sect. 3

the performance of the regional model is assessed in terms of its meteorological forecast and $CO_2$ simulation capability through comparisons with global model results. The benefit of higher horizontal resolution is investigated in section 4, followed by a discussion of the results and a conclusion in Section 5.

## 2 Methods and data

### 2.1 Model description

#### 2.1.1 GEM-MACH-GHG

GEM-MACH-GHG (Polavarapu et al., 2016) is a global GHG transport model, coupled with the meteorological model, wherein tracers are transported every time step. The horizontal resolution of the model is 0.9° using a global uniform latitude-longitude grid (400 × 200 grid points) and there are 80 vertical levels, spanning the surface to 0.1 hPa. For meteorology and tracer transport, a semi-Lagrangian advection scheme is used. Additionally, a global mass fixer was implemented for the

transport of tracers in order to conserve the global mass of $CO_2$ during model forecasts. Kain and Fritsch (Kain and Fritsch, 1990; Kain, 2004) was implemented for convective transport of tracers through the deep convection. More details about the model can be found in Polavarapu et al. (2016). While $CO_2$ is regarded as an inert trace gas in the model, methane ($CH_4$) and carbon monoxide (CO) utilise a simple parameterized climate chemistry. Specifically, the full troposphere chemistry package employed in GEM-MACH is replaced by simple hydroxide reactions related to oxidations of $CH_4$ and CO in the atmosphere,

along with the conversion of $CH_4$ to CO.

As the operational version of GEM is updated periodically, model parameters are invariably tuned to optimize the performance of the model. In the previous configuration of GEM-MACH-GHG used in Polavarapu et al. (2016), thermal eddy diffusivity values within the PBL calculated by GEM were overridden to enhance vertical mixing of $CO_2$ concentration. A minimum value of 10 $m^2$ $s^{-1}$ was imposed within the PBL to prevent too little vertical mixing of $CO_2$ in boreal summer because

low values resulted in spuriously low $CO_2$ concentrations on model levels near the surface in daytime when the magnitude of biospheric fluxes sinks is great (Polavarapu et al., 2016). In contrast, the lower limit imposed in the operational version of



GEM-MACH is 0.1 $m^2\ s^{-1}$ which was also empirically chosen for air quality applications. In this study, we use a more recent version of GEM which has better vertical mixing within PBL than the version used in the previous study. This improvement allowed us to revise the thermal eddy diffusivity minimum imposed with the PBL to 1 $m^2\ s^{-1}$ from 10 $m^2\ s^{-1}$ for all simulations (with both global and regional models) conducted in this study because the previous value resulted in too low $CO_2$

concentrations at model levels near the surface over snow covered regions, e.g. Alberta and Saskatchewan, in boreal winter. The impact of the revised value in summer daytime is minimal, and some improvements are found in night time, making the diurnal cycle of modelled $CO_2$ concentrations more realistic overall.

Results from the global model with 0.9° horizontal grid spacing are used as the reference experiment for the verification of the newly developed regional model. However, to provide lateral boundary conditions (LBC) of $CO_2$

concentration and meteorology to the regional model, a higher horizontal resolution of 0.45° (800 × 400 grid points) is needed to avoid numerical instability in meteorological forecasts caused by drastic change of spatial resolution at the lateral boundary of the regional model domain. The original configuration is somewhat coarse to be used as LBCs for our regional model with 10 km horizontal resolution. Therefore, we also run the global model with a 0.45º horizontal grid spacing. All other configurations except horizontal grid spacing are the same with those used in coarse (0.9°) resolution global model.

**2.1.2 Extension to regional domain**

For the regional model simulation, a rotated latitude-longitude map projection with approximately 10 km horizontal grid spacing and a hybrid vertical coordinate is used. The domain of the regional model covers most of Canada and the US, as shown in Fig. 1, and consists of 528 by 708 grid points. The number of vertical levels is the same as in the global model as described in section 2.1.1, namely, 80 levels spanning the atmosphere from the surface to 0.1 hPa. Since the number of grid

points is also almost 5 times greater than that used in the 0.9° global model, the new regional model is more expensive to run. The physics packages used in the regional model are similar to those of the global model and GEM-MACH, and include radiation (Li and Barker, 2005), boundary layer mixing (Bélair et al., 1999), shallow (Bélair et al., 2005) and deep convection (Kain and Fritsch, 1990; Kain, 2004), orographic gravity wave drag (McFarlane, 1987) and nonorographic gravity wave drag (Hines, 1997a, b) schemes. More details are provided in Mailhot et al. (1998).

For a simulation with tagged tracers to distinguish each component of $CO_2$, e.g. those associated with biospheric, ocean and fossil fuel fluxes etc., a transport model should have the ability to simulate consistent masses across different components. In other words, the mass of the total $CO_2$ field should exactly equal the sum of the tagged $CO_2$ species, both globally and locally. This property is also required for estimating surface fluxes through Bayesian synthesis inversion (e.g. Enting, 2002). As already described in Polavarapu et al. (2016), the semi-Lagrangian advection scheme implemented in GEM

alters mass slightly during model integration. The magnitude of the change for short range forecasts is negligible but this is not the case for the lengthy simulations of inert trace gases such as $CO_2$. To compensate for mass losses of tracers, a mass conservation scheme (Bermejo and Conde, 2002) and a shape preserving locally mass conserving scheme (Sørensen et al., 2013) were applied to tracer fields. At the lateral boundaries of the regional model domain, a mass restoration scheme (Aranami





et al. (2015) scheme) is applied. These schemes, however, can make mixing ratios across multiple tracers inconsistent since they correct for global mass changes in local regions where tracer gradients are large. Since each tagged component has rather different spatial structure and gradients from the total $CO_2$ field, the mass fixes made to the individual tagged variables need not be consistent with that made to the total field. As a result, the sum of each component may not equal the total $CO_2$

concentration field. To address this issue, the monotonicity and mass conservation schemes applied during the advection step are turned off in the regional model. The impact of the configuration on total mass of $CO_2$ within the regional model domain was compared with total mass of $CO_2$ from an experiment using mass related schemes turned on. The results show that there is no significant difference between two configurations in terms of total mass of $CO_2$ in the whole model domain as well as modelled $CO_2$ concentrations at the lowest model level in which most surface-based in-situ observation sites are located (not

shown). This occurs because the majority of tracers' mass are injected into the regional model domain through its lateral boundaries. The mass of $CO_2$ from the surface flux is small compared to the total mass of $CO_2$ in the atmosphere of the regional model domain and the signal of surface fluxes exits the lateral boundaries during model integration before they reach the upper levels of the atmosphere (e.g. upper troposphere and stratosphere). Therefore, in the regional model, we obtain perfect "additivity" of the tagged components with negligible loss of mass conservation.

**2.2 Surface flux**

In this study, the optimised $CO_2$ fluxes from NOAA's CarbonTracker, version CT2016 (Peters et al., 2007, with updates documented at http://carbontracker.noaa.gov) were used as surface $CO_2$ fluxes for $CO_2$ simulations. The temporal resolution of the surface flux is 3 h. Because of their ready availability and careful validation, many studies aimed at global to regional to urban scales have used optimized fluxes from CarbonTracker for forward $CO_2$ simulations (Houweling et al., 2010; Ballav

et al., 2012; Díaz-Isaac et al., 2014; Polavarapu et al., 2016; Li et al., 2017; Wu et al., 2018).

The original CT2016 flux product is available on 1° by 1° horizontal grid spacing. However, the global and regional models have different horizontal grid spacing. Thus, fluxes are re-gridded to GEM's grids with 0.9°, 0.45° and 10 km horizontal spacing grid, respectively, in a mass-conservative way. In addition, one more redistribution method is applied to re-gridded fluxes on the 10 km grid. This process applies a land-sea mask to the regional model grid in order to avoid unphysical modelled

$CO_2$ concentrations caused by the different behaviour of vertical mixing over land and water grid cells. Because coarse resolution fluxes do not contain all the information needed for high resolution grid cells, considering only the size of a grid cell in re-gridding is insufficient because it would lead to too low or high modelled $CO_2$ concentrations relative to observed $CO_2$ concentrations in regions of strong surface $CO_2$ fluxes. With respect to fossil fuel emissions, for example, dynamic consistency is one of the important factors in regional scale $CO_2$ concentration simulation, in particular along coastal margins

(Zhang et al., 2014). Hence, biospheric and fossil fuel flux components on water grid cells where the fraction of land is less than 30%, including lakes and oceans, are redistributed into land grid cells (within a radius of 30 grid points) in order to simulate realistic $CO_2$ concentrations along coastlines in the regional model domain while minimizing the impact of



redistributed surface fluxes on $CO_2$ simulation and while conserving the total mass of surface fluxes within the regional model domain.

### 2.3 Observations

Modelled $CO_2$ concentrations are verified against observations from ObsPack (Masarie et al., 2014) which is maintained and

provided by NOAA. For surface measurement sites, not all observations available in the regional model domain are used in the evaluation. The following selection criteria are applied: (1) sites were used to infer optimized $CO_2$ fluxes in CT2016. Thus, we can expect that optimized surface fluxes from CT2016 provide information about sources and sinks consistent with observed $CO_2$ concentrations at those sites so that differences in model simulation results may be attributable to model error, (2) sites have continuous measurements, (e.g. hourly data) as we want verify the results for all forecast hours, (3) sites have

no periods of missing data longer than one month (except for ESP (See Table 1 for full list of station abbreviations ) which has no data in January 2015) so results can be obtained for all seasons during experimental period. As a result, 19 measurement sites (11 sites in Canada and 8 tower sites in the US) are selected as shown in Fig 1 and listed in Table 1. For aircraft profiles of $CO_2$, measurement sites available over Canada and the US in the year 2015 were selected.

### 2.4 Experiment design

Three experiments are performed as listed in Table 2. GLB90 is the reference experiment using the global model with 0.9° horizontal grid spacing. GLB45 which uses the global model with 0.45° horizontal grid spacing is carried out to provide LBCs to the regional model. LAM is the regional model run with 10 km horizontal grid spacing. The simulation period is one year for 2015. In the analysis, the first 10-days of simulations are regarded as a spin-up period and discarded.

      Figure 2 depicts the global and regional model cycles. In each forecast, the weather forecast and $CO_2$ transport by

forecasted wind field are performed simultaneously in every time step. For the initial condition (IC) of meteorological fields for the two global models at the beginning of every cycle, the operational global analysis products from the global deterministic prediction system (GDPS; Buehner et al., 2015), whose horizontal resolution is roughly 25 km on a regular lat-long grid or, as of 15 December 2015, a yin-yang grid (Qaddouri and Lee, 2011) are used. The archived data are interpolated to our low-resolution grids and topographies of GLB90 and GLB45, separately. For the IC of meteorological fields for the regional model,

the operational regional analysis products from the regional deterministic prediction system (RDPS; Fillion et al., 2010; Caron et al., 2015) are used. The regional model grid is a subset of the model domain of the operational RDPS with the same horizontal resolution, sharing grid points on the same latitudes and longitudes. Therefore, it is not necessary to perform a horizontal interpolation at the start of every cycle. Also, a spin-up period for the meteorological forecast is unnecessary. Both operational global and regional meteorological analyses are produced 4 times per day with a 6 h assimilation window centred on the

analysis time. We only use analyses produced on 00:00 UTC every day as an IC of meteorology. Thus, a 24-hour weather forecast is produced during each 24 h $CO_2$ cycle and these forecasts are replaced by new analyses every 00:00 UTC, with the exception of microphysics tracers which are retained to allow a hot start for hydrometeor fields (Milbrandt et al., 2016). On



the other hand, the mass of $CO_2$ in a model grid volume is kept during cycles without replacements. The 24 h forecast of $CO_2$ from the previous cycle is used as the IC of the $CO_2$ field for the next cycle at 00:00 UTC. This is combined with the updated meteorological analysis for a complete initial state for the coupled model. Such 24 h forecast cycles are also used in other global model systems (e.g. Agustí-Panareda et al., 2014; Ott et al., 2015).

5       The IC of 3-D atmospheric $CO_2$ concentrations at the beginning of all three experiments is taken from CT2016 $CO_2$ concentrations at 00:00 UTC 1 January 2015. The LBC of $CO_2$ concentrations for the regional model are obtained from GLB45 and include hourly meteorological and $CO_2$ fields.

One more possible configuration is that of using operational RDPS forecasts as meteorological LBCs for the regional model, which is similar to the configuration of the operational regional GEM-MACH (Moran et al., 2010). We tested this

configuration and compared modelled $CO_2$ concentrations with the LAM experiment's modelled $CO_2$ concentrations. Negligible difference was found (not shown), therefore we decided not to include that configuration in this study because our purpose is in developing an integrated global/regional forward modelling framework for GHG simulation as shown in Fig. 2.

### 2.5 Sampling method and metrics

In order to evaluate the performance of $CO_2$ simulations and meteorological forecasts, a series of metrics are used as described

below. Modelled or forecasted values are sampled at the observed location by applying horizontal and vertical interpolation to model fields rather than selecting the nearest grid point to measurement locations, and selecting the time step closest to observed time. To sample modelled $CO_2$ concentrations, the sampling height above the ground level (or the model surface level) is considered to determine the altitude for vertical interpolation instead of using the actual sampling height above sea level (i.e. the sum of the altitude of an observation site plus intake height). Coarse horizontal resolution models cannot resolve

well the complex topography (e.g. mountain regions) around some measurement sites. As a result, the altitude of model topography may be far above or below the actual height of a measurement site. If the height of a model-sampled observation is erroneously placed in the PBL (free troposphere) as a result of coarse model topography, this can result in an unphysical, too strong (too weak) diurnal cycle of $CO_2$ compared to observed values (see Agustí-Panareda et al., 2019). From a comparison of the two sampling methods, it was found that the LAM experiment is not sensitive to the vertical sampling method, as

expected, because it can resolve actual topography well thanks to the higher spatial resolution, but the GLB90 experiment is sensitive to the method at a number of measurement sites (not shown). Thus, in order to reduce the topography mismatch problem in the coarse resolution global model and investigate the impact of higher horizontal resolution without this problem, the method using intake height is used to help the global model capture the behaviour of the PBL variations with time and height. A detailed discussion of the vertical sampling methods in connection with horizontal resolution is found in Agustí-

Panareda et al. (2019).

To analyse our results, including $CO_2$ and meteorology, bias and standard deviation of forecast error (STDE) are used. The bias is defined as





$$\text{Bias} = \bar{X} = \frac{1}{N}\sum_{i=1}^{N}(M_i - O_i), \qquad (1)$$

where $N$ indicates the number of observations, $M_i$ indicates modelled $CO_2$ concentration or meteorological forecast and $O_i$ indicates the corresponding observation.

The STDE is defined as

$$\text{STDE} = \sqrt{\frac{1}{N}\sum_{i=1}^{N}(X_i - \bar{X})^2}, \qquad (2)$$

where

$X_i = M_i - O_i$ and the overbar refers to the bias of the quantity.

To calculate the amplitude of $CO_2$ diurnal cycle (or other frequencies) for a measurement site or grid point, we use a Discrete Fourier Transform (DFT) technique. The linear trend in hourly $CO_2$ time series from a specific location is removed

first, then the DFT technique is applied to the detrended $CO_2$ time series to extract the amplitude of $CO_2$ variability across temporal scales, from synoptic to diurnal to sub-daily scales, as discussed in Section 4.4.

## 3 Model evaluation

### 3.1 Evaluation of meteorological fields

Before considering $CO_2$ simulation results, weather forecasts from the three experiments are verified against observations over

the regional model domain and are compared with each other. The motivation for doing this is two-fold: (1) to check that the meteorological forecasts from our regional model have not drifted away from the operational forecasts which have been produced and maintained by the CMC for many decades and (2) to compare the regional model results with the global model results. The first check is necessary because the configuration for weather prediction and the GEM model version used in this study are different from what was used to produce the operational forecast in 2015. For example, LBC in RDPS were obtained

from a global model forecast using a 33 km horizontal grid spacing (Caron et al., 2015) (or with a 25 km horizontal grid spacing as of 15 December 2015), with a different model domain extent and vertical coordinate. As shown by Polavarapu et al. (2016), the performance of the weather forecast by the global model is already well evaluated. The uncertainty of 24 h weather forecasts in the global model corresponding to GLB90 experiment in this study is comparable with those of reanalyses provided by three operational centres; monthly and zonal means of fields in 2009 and 2010 are within acceptable range on

global scales. Thus, in this section, we focus on the regional model's results and on differences between experiments on the regional domain.

Figure 3 shows the bias and STDE of 24 h forecast error of the three experiments for vertical levels from 1000 hPa to 10 hPa in July 2015. The same numbers of North American radiosonde observations are used in each of the three sets of verifications and these are indicated in the right of each panel. Statistical significance of the differences using a T-test for the

means and an F-test for the standard deviations with the 95% confidence level were computed but not shown explicitly. However, the discussion below uses this information in that we only mention results that are statistically significant. The three





experiments show good agreement with observations in terms of bias and STDE. For zonal wind, there are quite small differences among experiments and the scores remain within the range of operational forecasts, except at 925 and 850 hPa where biases in the GLB90 and GLB45 experiments are slightly better than those in the LAM experiment (Fig. 3a). For wind speed, unlike the zonal wind, forecasts in LAM experiment are better than those from the GLB90 experiment for levels from 5 925 hPa to 50 hPa and better than those in the GLB45 experiment for levels from 300 hPa to 70 hPa and at 700 hPa (Fig. 3b). For geopotential height, the forecasts in the LAM experiment are better than those in the GLB90 and GLB45 experiments from 400 hPa to 10 hPa where relatively large positive biases in GLB90 and GLB45 experiments exist. In addition, the STDE in the LAM experiment is better at all vertical levels except a few levels (Fig. 3c). For temperature, the forecasts in the LAM experiment are better than those in the GLB90 and GLB45 experiments from 1000 hPa to 70 hPa with the exception of 150 10 hPa (Fig. 3d).

The scores for December 2015 are shown in Fig. 4. The differences in bias and STDE among experiments are smaller than those in July. In addition, patterns in the reduction of bias and STDE from coarse horizontal resolution to higher horizontal resolution can be seen much more clearly than in July, which means that the values in the GLB45 experiment are located between those of the GLB90 and LAM experiments. For zonal wind, there are quite small differences in bias and STDE among 15 experiments as was the case in July (Fig. 4a). For wind speed, forecasts in the LAM experiment are better than those in the GLB90 experiment from 850 hPa to 150 hPa and those in the GLB 45 experiment at 500 hPa and 400 hPa, but not better at 925 hPa (Fig. 4b). For geopotential height, both bias and STDE in the LAM experiment are better than those in the GLB90 experiment at most pressure levels except from 700 hPa to 400 hPa, while the LAM experiment is better than GLB45 from 1000 hPa and 925 hPa (Fig. 4c). For temperature, the bias in the LAM experiment is better than that in the GLB90 and GLB45 20 experiments from 925 hPa to 250 hPa (Fig. 4d).

It is also worth considering how our meteorological forecasts compare to those of other systems. Agustí-Panareda et al. (2019) show RMSEs of vector wind for January and July 2014 from 1 d forecasts from the Copernicus Atmosphere Monitoring Service (CAMS). Our RMSE scores computed using the data from Figs. 3 and 4 for wind speed are shown in Table S1 and these can be compared to their Fig. 4. Our LAM scores are lower than those of the 9 km CAMS at all heights in January 25 and July. However, this is not a fair comparison since their scores are for global domain whereas we consider the North American domain, and their values are for 2014 but ours are for 2015. Nevertheless, the comparability of the scores further suggests that our LAM is performing well in terms of 24 h meteorological forecasts.

The number of available observations at 1000 hPa is much smaller than that of other pressure levels (see numbers in right side of each panel in Fig. 3 and 4) because typical altitudes of many sites are above the level corresponding to 1000 hPa 30 and surface pressures may be below 1000 hPa depending on the synoptic situation, there is little confidence in the verification at this level by means of radiosondes. A better approach to rigorously investigate the performance of weather forecasts at lower levels is to use surface observations because of their much greater numbers (in both space and time). Therefore, weather forecasts in the three experiments are also verified against observations near the surface. Figure 5 shows bias and STDE of sea level pressure, 2-m temperature and 10-m wind speed as well as the Heidke Skill Score (HSS) (Wilks, 2006) of 10-m wind



speed for July 2015. The STDE of sea level pressure in the LAM experiment is lower than those in the GLB90 and GLB45 experiments, while the bias of sea level pressure in the LAM experiment is slightly lower than those of the GLB90 and GLB45 experiments. However, the difference between the LAM and the GLB90 and GLB45 biases does not exceed 0.5 hPa (Fig. 5a). The STDEs of 2-m temperature and 10-m wind speed in the LAM experiment are smaller than those in GLB90 and GLB45

experiments for all forecast hours (Fig. 5b and c), which implies that the error of forecasts from the LAM experiment fluctuates less than those of the GLB90 and GLB45 experiments. Also, the better results of the LAM experiment in 10-m wind direction is evident in the higher HSS of the LAM experiment (Fig 5d). Higher HSS means a better forecast of wind direction. In addition, root-mean squared errors (RMSEs) of variables in the LAM experiment are lower than those of the GLB90 and GLB45 experiments (not shown).

In December 2015, the better forecasts in the LAM experiment compared to those of the GLB90 and GLB45 experiments can be seen more clearly (Fig. 6). The bias and STDE of each variable in the LAM experiment are lower than those of GLB90 and GLB45 experiments at most forecast hours (Fig. 6a-c), and higher HSS of 10-m wind direction are evident at all forecast hours (Fig. 6d).

In summary, the LAM experiment produces reasonable meteorological forecasts in comparison with meteorological

observations and better results relative to the GLB90 and GLB45 experiments, in particular at surface levels which are important for correctly capturing the flow of $CO_2$ affected by surface fluxes and boundary layer mixing, with reductions in both bias and STDE. Since forecasted meteorological fields are used to transport $CO_2$ in each simulation individually, better $CO_2$ simulations in the LAM experiment can be expected in the verification of modelled $CO_2$ concentrations.

### 3.2 Evaluation of $CO_2$ fields

The $CO_2$ fields in the LAM and other experiments are investigated in terms of monthly bias and STDE of daily afternoon (12:00-16:00 LST) modelled $CO_2$ concentrations at the measurement sites shown in Fig. 1 and listed in Table 1 (Fig. 7). CT2016 results are included as a reference in order to verify results of three experiment simultaneously. In general, bias and STDE in summer are larger than in other seasons at most sites except BAO, SCT, WGC and WKT. Better results in CT2016 than in all three experiments at many sites can be seen, especially in June to October. Surface $CO_2$ fluxes from CT2016 were

inferred by minimizing the difference between observations and forecasts of TM5 (Krol et al., 2005) which is the transport model used in CT2016, reflecting the signature of TM5's transport which may not match with GEM-MACH-GHG's transport (Polavarapu et al., 2016). As a result, larger biases in the three experiments relative to CT2016 at the above mentioned sites may be expected due to the discrepancies of modelled $CO_2$ concentrations between CT2016 and GEM-MACH-GHG. In contrast, the three models show similar biases to each other, except at ESP and WGC especially in boreal summer. Since the

LAM experiment uses LBCs of $CO_2$ concentrations from the GLB45 experiment, information about large scale transport of $CO_2$ concentration is reflected in LAM experiment.

Now we consider the seasonal variation of the performance of the LAM model over the regional (North American) domain. The seasonal bias and STDE of modelled $CO_2$ concentrations in the LAM experiment are shown in Fig. 8, based on





daily afternoon $CO_2$ concentrations. In boreal winter (DJF) and spring (MAM), there are mainly positive biases at most Canadian sites, while STDE are small relative to other sites in the US. The magnitude of surface $CO_2$ fluxes in those seasons over Canada is quite small (not shown), and thus bias at the Canadian sites contributes little to the overestimation of $CO_2$ concentrations. This result suggests that biases included implicitly in the LBC of $CO_2$ concentrations provided by the GLB45

experiment is more important in determining the biases in the regional model domain in those seasons. Four sites (BRA, EST, ETL and LLB) in Alberta and Saskatchewan provinces show relatively large STDE in DJF due to local influences of surface fluxes trapped within a shallow boundary layer by low temperatures. On the other hand, in boreal summer (JJA), large STDE can be seen with negative biases at most sites. The large biases and STDE in JJA may be attributed to errors in terrestrial $CO_2$ fluxes within the regional model domain. As shown in Fig. 7, the GLB90 and GLB45 experiments underestimate $CO_2$

concentrations over northern sites in JJA. That is also reflected in the underestimation of $CO_2$ concentrations in LAM experiment because the same surface fluxes are used in the simulations. Finally, in boreal autumn (SON), both biases and STDE show moderate values between MAM and JJA. Negative biases at northern sites in LAM experiment is partially due to biases in LBCs obtained from the GLB45 experiment. In summary, the performance of the regional model partially depends on biases in the global model which provides the LBC, and the relative importance of these biases varies with season. In this

regard, the use of CT2016 posterior fluxes to drive our global models exacerbates such biases. However, in the future when our global models provide their own flux estimates, such biases may be reduced. Furthermore, the need for a better understanding of the relative role of initial and boundary conditions and surface fluxes in controlling $CO_2$ distributions within the regional model's domain is evident. This is the subject of our future work.

## 4 The impact of horizontal resolution on $CO_2$ simulation

Since our regional model requires more computational resources (due to the greater number of grid points and shorter time step) than our global (GLB90) model, it is important to consider the added benefit of the higher horizontal resolution on $CO_2$ simulations. In this section, modelled $CO_2$ concentrations from the three experiments are analysed from the perspective of spatial patterns and vertical profiles of $CO_2$ concentrations and the reproducibility of temporal patterns against atmospheric $CO_2$ observations.

### 4.1 Spatial patterns of surface $CO_2$ concentrations

Biases between two experiments are compared pairwise for four seasons (Fig. 9). Three comparisons are shown because three experiments are conducted. Since bias can have both positive and negative values, absolute bias is used in the calculation. Blue (red) colour means the higher horizontal resolution model simulated smaller (larger) absolute bias compared to coarser one. In DJF, the GLB45 and LAM experiments are better than the GLB90 experiment. However, the LAM experiment is not better

than GLB45 experiment at U.S. sites except WGC and BAO. In MAM, the differences among the experiments are the smallest, except at ESP near the west coast of North America. Differences between the LAM and GLB45 experiments at northern





Canadian sites are quite small in DJF and MAM, which is associated with weak surface $CO_2$ fluxes in those seasons. In JJA, the reduction in bias resulting from the higher horizontal resolution model can be seen clearly and the magnitude of reduction is higher. The LAM experiment shows better results than both the GLB90 and GLB45 experiments at most sites. On the other hand, the GLB45 experiment is not better than the GLB90 experiment except at a few sites. This nonlinearity of the

improvement with increased resolution is consistent with the results of Agustí-Panareda et al. (2019) although their conclusions are based on RMSE rather than absolute bias. In SON, the results are similar to those in JJA, namely, the LAM experiment is better than both the GLB90 and GLB45 experiments at most sites except ESP. At many sites, the benefit of finer grid spacing is evident, but higher horizontal grid spacing does not always guarantee a lower magnitude of bias for all sites.

Figure 10 shows the differences of STDE between two experiments. The spatial pattern of differences of STDE is

different from that of bias. More blue dots are evident indicating that the STDE of higher horizontal resolution model is smaller than that of the coarser horizontal resolution model at most sites. Specifically, the LAM experiment shows better results than both the GLB90 and GLB45 experiments in DJF. As shown in Fig. 6c and d, better forecasts of 10-m wind speed and direction at screen level in the LAM experiment should help to reduce STDE in LAM experiment relative to the two global model experiments. In MAM, the impact of horizontal resolution is very small at most sites except near the southern boundary of the

regional model domain due to the weak magnitude of surface $CO_2$ flux in this season. The ratio between $CO_2$ concentrations resulting from surface $CO_2$ fluxes within the regional model domain and background $CO_2$ concentrations from the GLB45 experiment shows that the contribution of surface $CO_2$ flux is least in MAM due to the small magnitude of surface $CO_2$ flux (not shown). In JJA, on the other hand, the magnitude of the difference is larger than in other seasons in both positive and negative directions. Finally, in SON, the improvement due to finer grid spacing can be seen.

In summary, the difference of bias and STDE between experiments provide evidence of improvement in $CO_2$ simulations due to finer horizontal resolution and better wind forecasts near the surface in the LAM experiment. The pattern of the differences are strongly associated with the spatial and seasonal patterns of the magnitude of surface $CO_2$ fluxes used in simulations.

**4.2 Vertical profile of $CO_2$ concentrations**

We now consider the quality of modelled $CO_2$ concentrations in the free troposphere. Observed profiles of $CO_2$ can reveal the signatures of vertical mixing, so they can be used to measure the performance of transport models (Lin et al., 2006). The seasonal bias and STDE of vertical profiles of modelled $CO_2$ concentrations against NOAA aircraft profiles (Sweeney et al., 2015) over sites in Canada and the US inside the regional model domain are shown in Fig. 11. Modelled $CO_2$ concentrations are sampled at the exact location and height of observations by applying vertical and horizontal interpolation to 3-D model

fields at a time step close to observed time. Then, averages over all profiles of modelled and observed values for a season are binned into 1 km thick layers. The three experiments generally overestimate $CO_2$ concentrations in DJF (at 1000 m) and MAM and underestimate them in JJA and SON, which is consistent with the comparisons against surface $CO_2$ measurement sites shown in Fig. 8. The magnitude of the bias does not exceed about 2 ppm in any altitude or season, and it decreases with





altitude. Profiles of $CO_2$ concentrations near the ground are difficult to simulate due to the strong influence of surface fluxes (Geels et al., 2007). The range of the biases in the three experiments are similar to that seen in our previous study (Polavarapu et al., 2016) as is the direction of the biases. Specifically, the bias changes sign with height, with positive biases at low altitudes and negative biases in DJF and MAM.  In JJA and SON, the bias remains negative at almost all heights. The LAM experiment

generally has the smallest biases for all seasons and altitudes, in particular, below 4000 m in JJA when the influence of surface $CO_2$ fluxes is significant through active vertical mixing. Lower wind speeds in boreal summer compared to other seasons causes accumulation of surface fluxes over North America in the lower 4000 m (Sweeney et al., 2015). Reduced bias and STDE of forecasted temperature profiles in the LAM experiment in July (Fig. 3d) may help to improve vertical advection in the LAM simulations relative to the global model experiments through improved buoyancy calculations. This may explain

why the LAM experiment has a better ability to simulate vertical profiles of $CO_2$.

### 4.3 Temporal patterns

We evaluate modelled $CO_2$ concentrations at various temporal scales including synoptic variability and the diurnal cycle. First, synoptic variability of modelled $CO_2$ concentrations is analysed. Figure 12 shows Taylor diagrams (Taylor, 2011) of modelled $CO_2$ concentrations in the afternoon compared with observations. Since the domain of the LAM experiment covers a variety

of geographic regions across Canada and the US, including mountain, continental and coastal sites, the synoptic variability of $CO_2$ is not expected to be captured well at all sites. In DJF, the variability of modelled $CO_2$ concentrations in the LAM experiment is closer to the observed variability than that captured in the GLB90 and GLB45 experiments in accordance with decreased STDE seen in Fig. 10 (Fig. 12a). In MAM, the variability of modelled $CO_2$ is scattered with relatively lower correlations than other seasons (Fig. 12b). In general, due to the onset of growing season in MAM, transport models tend to

produce lower correlations with observed $CO_2$ (Geels et al., 2004; Pillai et al., 2011; Agustí-Panareda et al., 2014). In JJA, despite having larger biases than in other seasons (Fig. 7), correlations are quite reasonable lying mostly between 0.6 and 0.95 (Fig. 12c). However, the variability in the $CO_2$ concentrations tend to be overestimated. This could be mainly due to the large uncertainty in biospheric fluxes (Patra et al., 2008). Also, the range of correlations is the biggest -- between approximately 0 and 0.95. In SON, the synoptic variability of $CO_2$ is well captured by all experiments (Fig. 12d). Many sites have correlations

higher than 0.9, standard deviations similar to observed variability, and the least normalised RMSE (the distance from the reference point on the x axis) relative to other seasons. We expect the LAM experiment to produce higher correlations and smaller normalised RMSEs, and normalised standard deviations approaching 1. Indeed, the LAM experiment tends to simulate well the observed variability of $CO_2$ and it produces smaller normalised RMSE relative to the GLB90 and GLB45 experiments to some extent although the results vary according to site, and each experiment shows similar seasonal patterns which are

driven by the weather forecasts and surface fluxes.

Thus far, modelled $CO_2$ concentrations in the afternoon time (12:00-16:00 LST) have been analysed. Henceforth, data at all times of the day and night are retained. Figure 13 shows the mean diurnal cycle of modelled and observed $CO_2$ concentrations for July and December at WGC where the most significant differences among the three experiments are





observed. In general, the three experiments simulate similar $CO_2$ diurnal cycles for other sites (not shown). Three sampling levels data are available at WGC in 2015. CT2016 is included as well for comparison purposes, but only results at the highest sampling level are shown because only observations at this level were used in the inversion in CT2016. The LAM experiment captures the $CO_2$ diurnal cycle well, but the GLB90 and GLB45 experiments and CT2016 do not, especially in July (Fig. 13a,

c and e). At the sampling level of 483 m, the GLB90 experiment overestimates morning time $CO_2$ concentrations and CT2016 overestimates night time $CO_2$ concentrations in July, while the GLB45 and LAM experiments capture the diurnal cycle (Fig. 13a) relatively well. This level (483 m) has a comparatively weak diurnal cycle because it is mostly decoupled from the surface at night and daytime enhancements are significantly diluted relative to lower levels. At lower sampling levels, 91 m and 30 m, both the GLB90 and GLB45 experiments overestimate night time $CO_2$ concentrations in July, whereas the LAM experiment

captures both day and night time $CO_2$ concentrations well (Fig. 13 c and e). This greater sensitivity to model resolution at night was also seen by Agustí-Panareda et al. (2019). WGC is located in a valley between two mountain ranges. The model topographies of GLB90 and GLB45 do not resolve this geography well due to their coarse horizontal resolutions. In contrast, the LAM experiment resolves the actual topography around the WGC site well relative to the two global models. In daytime, $CO_2$ concentrations are well simulated in the LAM and GLB45 experiments due to the strong vertical mixing (Fig. 13a, c and

e). In contrast, accumulated $CO_2$ in night time still remains in the afternoon time in the GLB90 experiment, leading to an overestimation of $CO_2$ in the afternoon at all sampling levels. In December, the LAM experiment simulates slightly better $CO_2$ concentrations and its standard deviation at all sampling levels, while the GLB90 and GLB45 experiments underestimate $CO_2$ concentrations (Fig. 13b, d and f).

In order to analyse $CO_2$ time series across various temporal scales beyond the diurnal cycle, the DFT method

explained in Section 2.6 is applied to hourly $CO_2$ time series. Figure 14 shows the amplitude of hourly $CO_2$ concentration time series across different temporal scales from 2 h to 92 days for the period from June to August 2015 at the LEF and WGC sites. Unfortunately, not all sites have hourly observations without missing values for the year 2015. These two sites have hourly data available for three months from June to August 2015 without missing values and, fortunately, reveal different properties. Thus, they were selected to illustrate the impact of increased horizontal resolution on $CO_2$ simulations on the time scales

captured by the models. At LEF, one sampling level, 396 m, satisfies our constraint of no missing data, and, at WGC, two sampling levels, 483 m and 91m, meet this constraint. At the LEF site, the three experiments capture well the signals across all temporal scales in observed $CO_2$ time series, including synoptic and diurnal variations (Fig. 14a and b). The topography mismatch of the GLB90 and GLB45 experiments are relatively small around at the LEF site. The intake height of measurements at LEF is 396 m above the ground at which laminar flow is more dominant than turbulent flow in night time so

that the respiration signal from surface does not reach the free troposphere and synoptic variability is more dominant (Davis et al., 2003; Wang et al., 2007). As a result, the differences of amplitude in the three experiments are less than about 0.8 ppm for all temporal periods (Fig 14b). At the WGC site, on the other hand, as already shown in Fig. 13, the GLB90 experiment simulates too strong diurnal cycles of $CO_2$ at lower sampling heights in July, which can be seen clearly as well in Fig 14c-f, with largest overestimation at the low sampling level (Fig. 14 e and f). However, it is not just the diurnal cycle of $CO_2$ from



June to August that is overestimated in the GLB90 experiment. Periods from sub-diurnal to longer day periods are also overestimated (Fig. 14c-f). Furthermore, while GLB45 performs better than GBL90, it also overestimates the diurnal cycle amplitudes and longer time scales at 483 m and 91m. Hence, the larger mismatch of topography results not only in inaccurate daily time scales but also other scales such as synoptic scales longer than 4-days.

Because the observation network used in this study is rather sparse, we also compute the amplitude of the diurnal cycle of $CO_2$ in model space instead of observation space to illustrate its spatial variability. Figure 15 shows the amplitude the diurnal cycle of $CO_2$ (i.e. 1-day period) over land regions in western and eastern North America using the same method used in Fig. 14. Hourly modelled $CO_2$ concentrations at the lowest model levels from the three experiments during JJA are selected. Results over oceans and northern Canada are excluded as these regions have much smaller amplitudes relative to those shown

in the figure. Even though the same coarse resolution surface $CO_2$ fluxes (originally on a $1° × 1°$ grid) are used in $CO_2$ simulations for all three experiments, the LAM experiment produces a more detailed spatial pattern than either of the GLB90 and GLB45 experiments on account of its better representation of topography and the variability of simulated PBL height (which is related to the vertical mixing of tracers within PBL, in particular over the Rocky, Sierra Nevada and Appalachia mountains, where terrain-induced circulation also plays an important role (De Wekker and Kossmann, 2015), and along

coastlines). The spatial pattern of the diurnal cycle is strongly correlated with that of surface biospheric fluxes (not shown) and model topography (Fig. 1). On the other hand, the impact of fossil fuel fluxes on $CO_2$ diurnal cycle is small because it has a consistently positive value although prescribed fossil fuel fluxes included in optimized flux from CT2016 have day of week and diurnal variations. Figure 15 also suggests that better representation of the diurnal cycle of $CO_2$ concentrations in the LAM experiment extends beyond the WGC site that was shown as an example in Fig. 13. Unfortunately, due to the limited

observation coverage, it is not possible to verify this pattern over the whole domain using observations, at the moment. Nevertheless, from Figure 15, it can be concluded that higher horizontal resolution might help to enhance the performance of $CO_2$ simulations even without using fluxes on a finer grid spacing. This is the case when comparing the $CO_2$ diurnal cycles between the GLB90 and GLB45 experiments. This hypothesis is consistent with the recent finding that higher horizontal resolution global models can simulate more detailed spatial patterns of the $CO_2$ diurnal cycle compared to low horizontal

resolution global transport models (e.g. Agustí-Panareda et al., 2019).

## 5 Discussion and conclusions

We have developed a regional atmospheric transport model for GHG gas simulation, as an extension of GEM-MACH-GHG which is ECCC's global atmospheric transport model for GHG simulation. The regional model shares much of the configuration of the global model, while its model domain is focused on Canada and the U.S. One gain from using the same

vertical coordinate in both the regional and global models is that there is consistency at lateral boundaries of the regional model domain. $CO_2$ simulations using the same surface $CO_2$ fluxes from CT2016 are performed with three configurations of 2 global models and 1 regional model in order to assess whether the newly developed regional model is working properly and to assess



the benefit of the regional model over the global model in terms of weather forecasts and $CO_2$ simulations. In a given experiment, a series of 24 h forecasts are replaced by operational analyses every cycle and used to transport $CO_2$ every time step, whereas transported $CO_2$ fields are not replaced but are kept during each 1-year simulations.

Meteorological forecasts in three experiments are verified against North American radiosondes and surface observations at screen level. All experiments show acceptable ranges of bias and STDE compared to observations. Overall, meteorological forecasts in the regional model show better results than both global models, especially in wind speed and direction at screen level which are of particular importance for $CO_2$ transport near the surface. We demonstrate the improvement of weather forecasts with increasing of horizontal resolution, which is most apparent in boreal winter. In addition, good quality meteorological forecasts in the global model are also required for providing meteorological LBCs to the regional

model with reduced errors at large scales. Indeed, the GLB45 experiment can provide good quality of meteorological LBCs to the regional model every hour which is more frequent than when using reanalyses that are available at 3 h or 6 h intervals.

While the meteorological forecasts from the higher resolution region model are demonstrably better than those of the coarser resolution global models, demonstrating improved $CO_2$ simulations with higher resolution is more challenging. For example, the impact of biases in the LBCs provided by the GLB45 experiment on $CO_2$ simulations near the Arctic region in

the regional model is large, especially in boreal spring. In a regional scale inverse modelling system, estimated fluxes within the regional model domain are strongly influenced by the inflow of $CO_2$ from the global transport model through lateral boundary (Schuh et al., 2010). Because LBCs of $CO_2$ include information of sources and sinks outside of the regional model domain, correct information at the lateral boundary is important to determine the sources and sinks in the regional model domain (Gerbig et al., 2003). As discussed in Polavarapu et al. (2016), GEM has different transport behaviour from the

transport model used in CarbonTracker, in particular over the Arctic region, as seen in time series of $CO_2$ concentrations and column-averaged $CO_2$. Thus, our models are not expected to perform better than CT2016 because we use surface $CO_2$ fluxes inferred by an inversion framework using a different transport model which has different transport behaviour. That is why our focus in this work is in the comparison of our regional and global models. However, despite this handicap, beyond afternoon time, we are able to find some benefits of our regional model over CT2016 when looking at the diurnal cycle of $CO_2$

concentrations at particular sites in which large topography mismatches exist, e.g., WGC. CT2016 and our global models did not capture diurnal cycles well, while our regional model did. This is a promising result because it suggests that using night time data in an inversion to estimate night time fluxes (e.g. Lauvaux et al., 2008) may be beneficial if a high resolution model is used. Currently, a GHG state estimation system using GEM-MACH-GHG and ECCC's operational Ensemble Kalman filter data assimilation system (Houtekamer et al., 2014) is under development. When posterior fluxes become available from our

global model, this will alleviate the issue of model transport error mismatches being embedded in the surface fluxes.

The regional model produces lower STDEs of $CO_2$ at surface measurement sites, in line with its lower STDE of meteorological forecasts. With respect to aircraft $CO_2$ profile comparisons, clear improvement of profiles of modelled $CO_2$ in the LAM experiment occurs at altitudes lower than 4000 m in boreal summer. Although the regional model domain is vast enough to include most of Canada and the U.S. so as to be able to estimate national to provincial scale surface GHG fluxes at





finer spatial resolution via inverse modelling in the future, it is not easy to obtain better results everywhere. For example, at the ESP site located on the coastline of Vancouver Island, British Columbia, the LAM experiment does not have a lower bias of modelled $CO_2$ than the GLB90 and GLB45 experiments in MAM and SON. Nonetheless, the overall performance of $CO_2$ simulations by the regional model is better than our global models. It is well known that only afternoon time $CO_2$ concentrations

are typically used in inversions due to the difficulty in capturing boundary layer evolution in most global transport models (Law et al., 2008; Patra et al., 2008). Significant improvement in reproducing the $CO_2$ diurnal cycle can be seen at sites located in complex terrain region. Reduced topographic mismatch in the finer horizontal resolution model is the major driving force behind reduced sampling and representation error. This effect is not limited to just the diurnal cycle but also occurs for synoptic variability of $CO_2$ at the level where large scale motions are dominant, and even more so at lower sampling levels near the

surface. In addition, the potential benefit in reproducing detailed diurnal cycle over regions with complex terrains hypothesized here is consistent with the findings of Agustí-Panareda et al. (2019).

Previous studies comparing high and low horizontal resolution transport models for $CO_2$ simulations concluded that some advantages can be attained by using higher horizontal resolution (Geels et al., 2007; Pillai et al., 2010; Díaz-Isaac et al., 2014). For example, better resolved amplitude and phase of short-term variability of $CO_2$ (Geels et al., 2007), reduced

representation errors (Pillai et al., 2010) and smaller-scale structures of modelled $CO_2$ that are more sensitive to the distribution of $CO_2$ fluxes (Díaz-Isaac et al., 2014) were attained by using higher spatial resolution of transport model. Indeed, we also find similar results as mentioned above, but these advantages from the regional model experiment are not obtained at every observation site (Fig. 9 and 10). Basically, increasing horizontal resolution gives some positive impact to some extent but it generally has a mixed impact in this study. Part of the reason may be due to the fact that our models are variants of the same

model but with different grid spacing and/or domain. Furthermore, the same coarse resolution surface fluxes were used with all models and this limits the potential for improvement (Remaud et al., 2018). In addition, the global model configurations used in this study already have relatively higher horizontal resolutions (0.9° and 0.45°) compared to other coarse resolution global transport models (e.g. Geels et al., 2007) and they all use the same number of (80) vertical levels as the regional model. Another major difference is that our global model is not an offline transport model which generally uses reanalyses as a

meteorological driver for transport. Instead we take advantage of operational analyses to initialize weather forecasts every day and produce weather forecasts at every model time step. A major limitation in validating the overall improved ability to capture fine spatial scales may simply be due to the current sparsity of verifying observations of $CO_2$. With vastly greater numbers of verifying observations, the meteorological simulations are demonstrably better with increased resolution. Since the regional model can better simulate the spatial heterogeneity of the diurnal cycle of $CO_2$ in model space (Fig. 15), better observational

density is needed to distinguish the performance of $CO_2$ simulations in the regional model in more detail.

A limitation of this study is the use of coarse resolution surface $CO_2$ fluxes in conjunction with the fine horizontal grid spacing of the regional model. For better simulation of $CO_2$, not only high-quality meteorological forcing but also high resolution prescribed surface fluxes are demanded (Locatelli et al., 2015). Higher spatial and temporal resolution fluxes could lead to better simulation of $CO_2$ concentrations (Lin et al., 2018) if the fluxes have correct space and time information about





the distribution of sources and sinks of $CO_2$ fluxes. The challenge is in obtaining high spatial and temporal resolution surface fluxes that are accurate. Preliminary investigations with a high resolution anthropogenic flux product revealed improved comparisons to observations at some sites but degradation at other sites. For that reason, we chose to start of investigation of the regional model by using fluxes with the same resolution as the global model and limiting the potential benefit of high

resolution to improved meteorological depictions.

The LBCs of $CO_2$ from the global model plays an important role, as shown in Fig. 8, dominating the bias in the regional model when the magnitude of surface flux is weak. In addition, the LBCs of meteorology also play an important role in $CO_2$ simulations. Thus there is a need to better understand the relative importance of initial conditions, boundary conditions and surface fluxes on the performance of the regional model in order to better characterize these components of $CO_2$ model

error within the regional domain. Indeed, the predictability of $CO_2$ on the regional domain and the relative role of initial and boundary conditions and surface fluxes on model error is a topic that is currently under investigation.

There are a number of extensions to this work that are envisioned. For example, the newly developed regional model is not limited to $CO_2$ simulations but also includes other greenhouse gases such as $CH_4$. Thus, a separate validation of the regional model's ability to simulate $CH_4$ is planned. The regional model can also be utilized to provide information (e.g. IC

and LBC) to urban scale forward or inverse modelling systems (e.g. Pugliese et al., 2018; Ishizawa et al., 2019). Lastly, and most importantly, an inverse modelling system for estimating surface $CO_2$ fluxes is being developed using the new regional GHG transport model to better understand the carbon cycle in Canada at finer spatial and temporal scales.

*Code and data availability.* The GEM-MACH-GHG model source code is publicly available at

https://zenodo.org/record/3246556 (Neish et al., 2019) under the GNU Lesser General Public License version 2.1 (LGPL v2.1) or ECCC's Atmospheric Science and Technology license version 3. The model output data are available at http://crd-data-donnees-rdc.ec.gc.ca/CCMR/pub/2019_Kim_GMD_Canadian_atmospheric_transport_model_for_simulating_greenhouse_gas_evolution_on_regional_scales/.

*Author contribution.* JK and MN developed model code. JK designed and carried out the experiments. All authors participated in the analysis of the results. The manuscript was prepared with contributions from all authors.

*Acknowledgements.* We are grateful to Colm Sweeney (NOAA ESRL) for providing the NOAA aircraft profiles and to Ken

Masarie of the NOAA Global Monitoring Division in Boulder, Colorado for compiling ObsPack. The National Oceanic and Atmospheric Administration (NOAA) North American Carbon Program has funded NOAA/ESRL Global Greenhouse Gas Reference Network Aircraft program. CarbonTracker CT2016 results were provided by NOAA ESRL, Boulder, Colorado, USA, from the website at http://carbontracker.noaa.gov. The ObsPack data (obspack_co2_1_GLOBALVIEWplus_v3.1_2017_10_18) were obtained for the period 2015 from





http://dx.doi.org/10.15138/G3T055. We would like to thank Doug Worthy of Atmospheric Science and Technology Directorate, Environment and Climate Change Canada, for developing and maintaining ECCC's greenhouse gas measurement network and for providing the $CO_2$ concentration measurement data. We thank Marc L. Fischer for useful comments on the manuscript. Data collection at the WGC site was partially supported by the California Air Resources Board through work at the Lawrence Berkeley National Laboratory, operating under U.S. Department of Energy under Contract No. DE-AC02-05CH11231. We thank Monique Tanguay and Felix Vogel for their careful internal review.

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





Table 1: Information of surface in-situ measurement sites used in this study.

|   | Code | Station name | Latitude (°) | Longitude (°) | Altitude (m a.s.l.) |
|---|------|-------------|-------------|--------------|---------------------|
| 1 | AMT | Argyle, Maine | 45.0345 | -68.6821 | 53 |
| 2 | BAO | Boulder Atmospheric Observatory | 45.03 | -105.004 | 1584 |
| 3 | BCK | Behchoko | 62.8 | -115.92 | 160 |
| 4 | BRA | Bratts Lake | 50.2 | -104.71 | 595 |
| 5 | CBY | Cambridge Bay | 69.13 | -105.06 | 35 |
| 6 | CPS | Chapais | 49.82 | -74.98 | 381 |
| 7 | EGB | Egbert | 44.23 | -79.78 | 251 |
| 8 | ESP | Estevan Point | 49.38 | -126.54 | 7 |
| 9 | EST | Esther | 51.67 | -110.21 | 707 |
| 10 | ETL | East Trout Lake | 54.35 | -104.99 | 493 |
| 11 | FSD | Fraserdale | 49.88 | -81.57 | 210 |
| 12 | INU | Inuvik | 68.32 | -133.53 | 113 |
| 13 | LEF | Park Falls, Wisconsin | 45.95 | -90.27 | 472 |
| 14 | LLB | Lac La Biche | 54.95 | -112.47 | 540 |
| 15 | SCT | Beech Island | 33.41 | -81.83 | 115 |
| 16 | SNP | Shenandoah National Park | 38.62 | -78.35 | 1008 |
| 17 | WBI | West Branch | 41.72 | -91.35 | 242 |
| 18 | WGC | Walnut Grove | 38.27 | -121.49 | 0 |
| 19 | WKT | Moody | 31.31 | -97.33 | 251 |

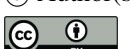



Table 2. Experiment design

| Experiment name | Horizontal grid spacing | Lateral boundary condition | Initial condition of meteorological fields | Time step |
|---|---|---|---|---|
| GLB90 | 0.9° (~ 90 km) | N/A | Global operational analysis | 15 min |
| GLB45 | 0.45° (~ 45 km) | N/A | Global operational analysis | 15 min |
| LAM | 0.09° (~10 km) | GLB45 experiment | Regional operational analysis | 5 min |





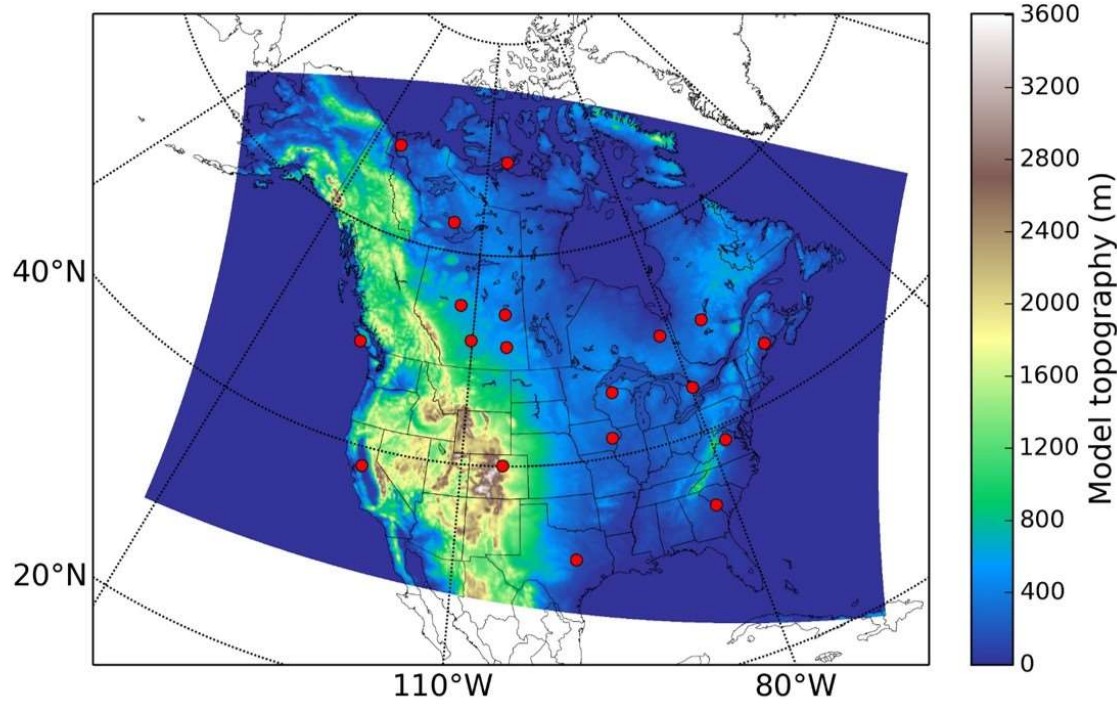

**Figure 1: Model topography of the regional model with 10 km horizontal grid spacing and $CO_2$ measurement sites used in this study (red dot).**



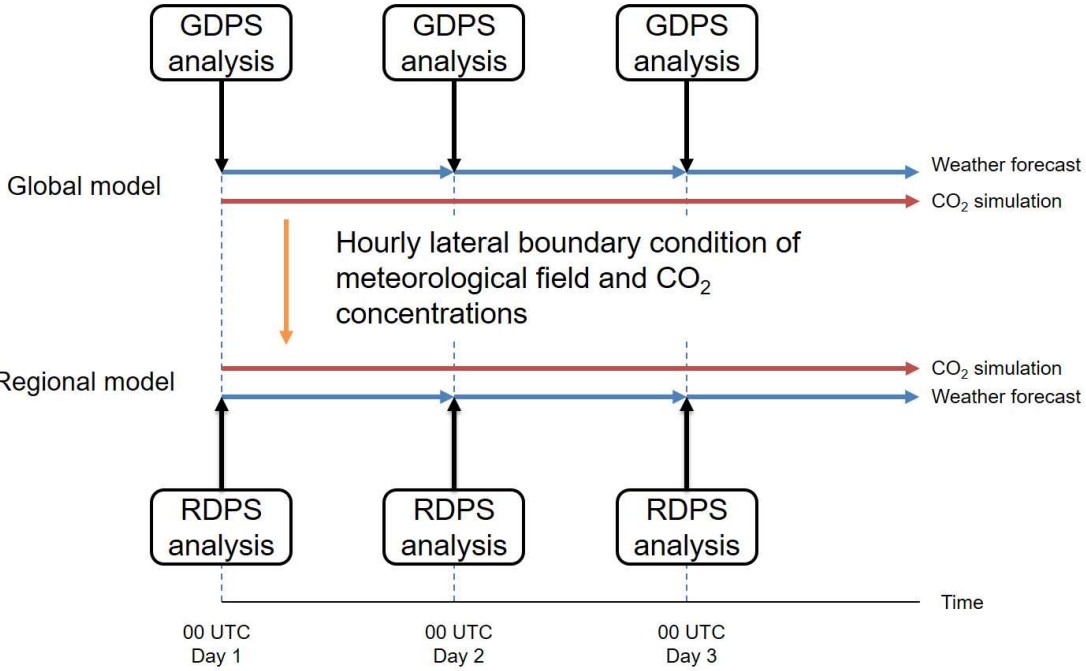

**Figure 2: Schematic diagram of GEM-MACH-GHG global and regional forward model cycles. Meteorological analyses are from CMC's operational global deterministic prediction system (GDPS) and regional deterministic prediction system (RDPS). Global and regional 24 h weather forecasts start at 00:00 UTC of each day with operational analyses, while CO₂ concentrations are kept during**

5  **cycles. The global forward model provides lateral boundary condition of meteorological and CO₂ fields to the regional forward model every hour.**



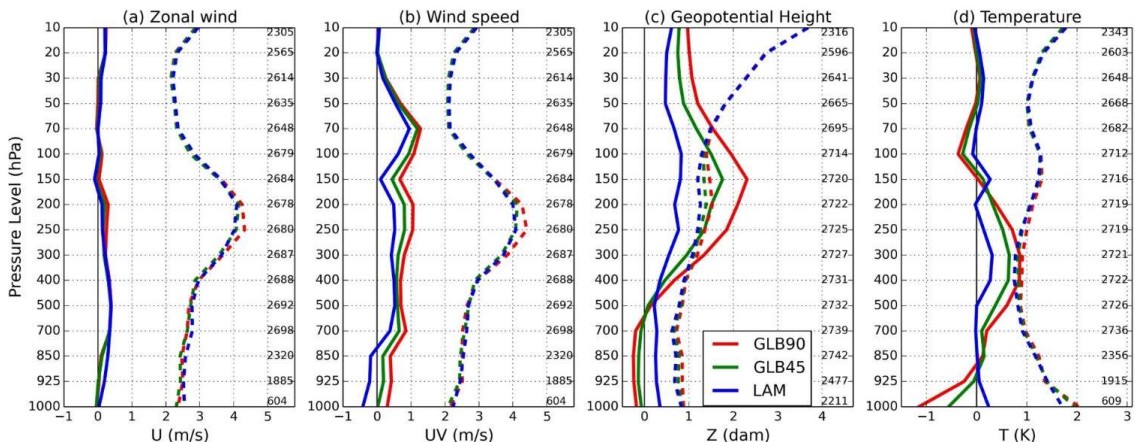

**Figure 3: The bias (solid line) and standard error (dashed line) of (a) zonal wind (m s⁻¹), (b) wind speed (m s⁻¹), (c) geopotential height (dam) and (d) temperature (K) from GLB90 (red), GLB45 (green) and LAM (blue) experiments, based on comparison 24-h forecasts against North American radiosondes for July 2015. Numbers on the left side of each panel denote the pressure level. Numbers on the right side of each panel denote the number of observations used in statistics at each pressure level.**



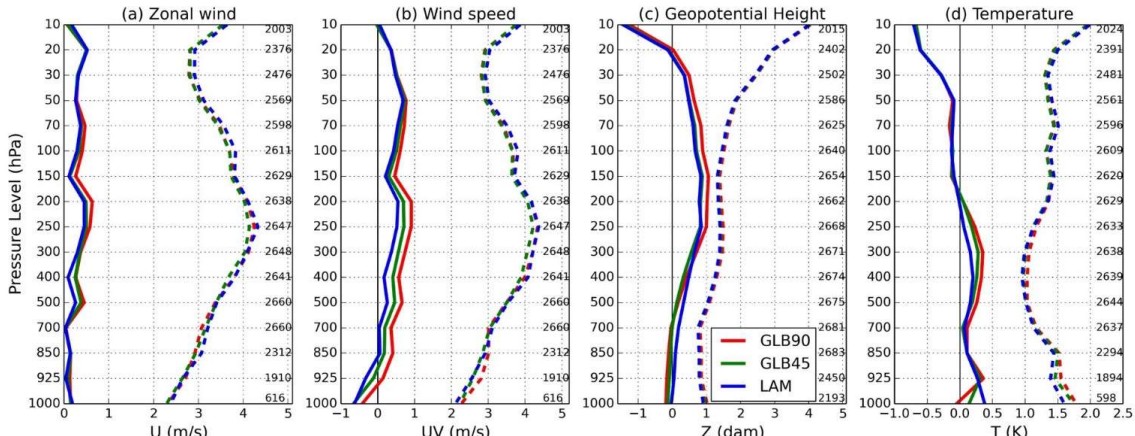

**Figure 4: As in Fig. 3, but for December 2015.**



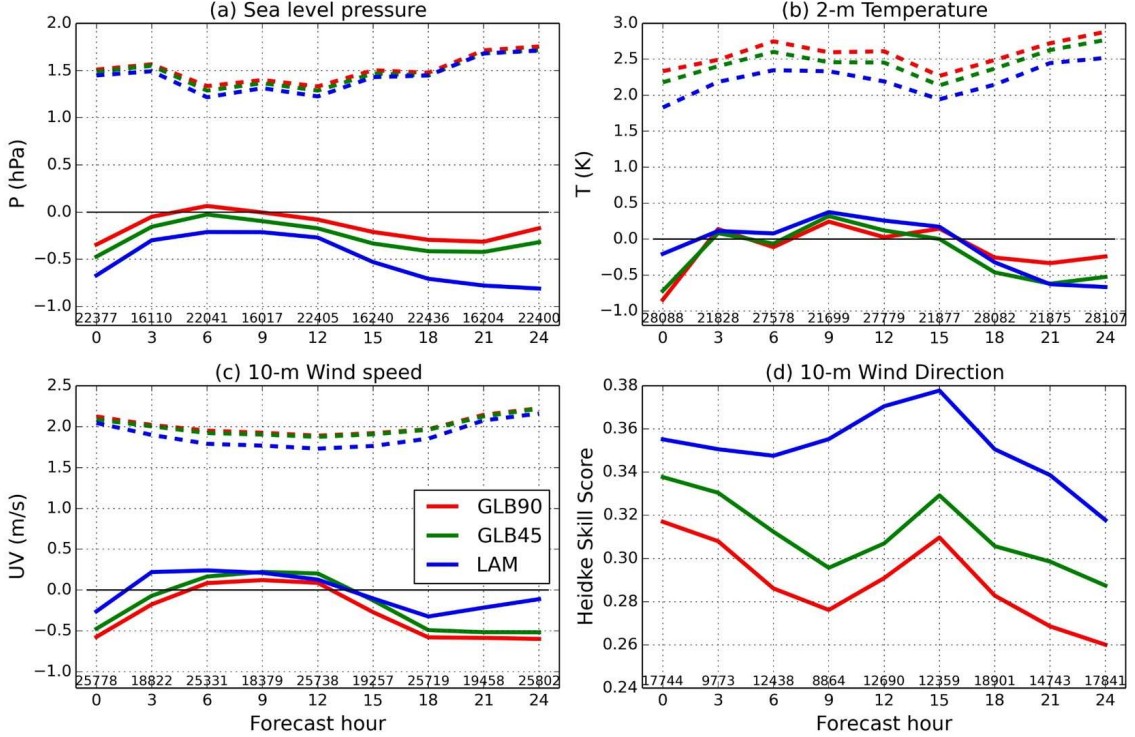

**Figure 5: The Bias (solid line) and standard error (dashed line) of (a) sea level pressure (hPa), (b) 2-m temperature (K), (c) 10-m wind speed (m s⁻¹) and (d) Heidke skill score of 10-m wind direction from GLB90 (red), GLB45 (green) and LAM (blue) experiments, based on comparison forecasts against surface-based stations over North America for July 2015. Numbers on bottom of each panel denote the number of observations used in statistics at each forecast hour.**



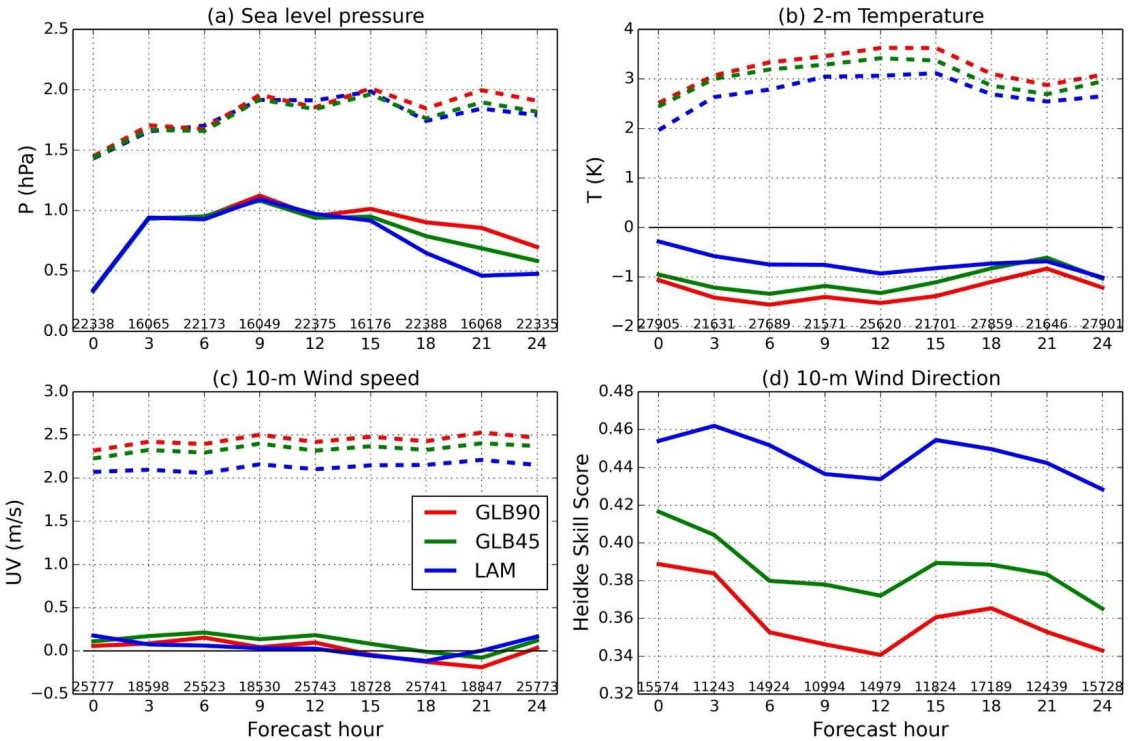

Figure 6: Same as Fig. 5, but for December 2015.

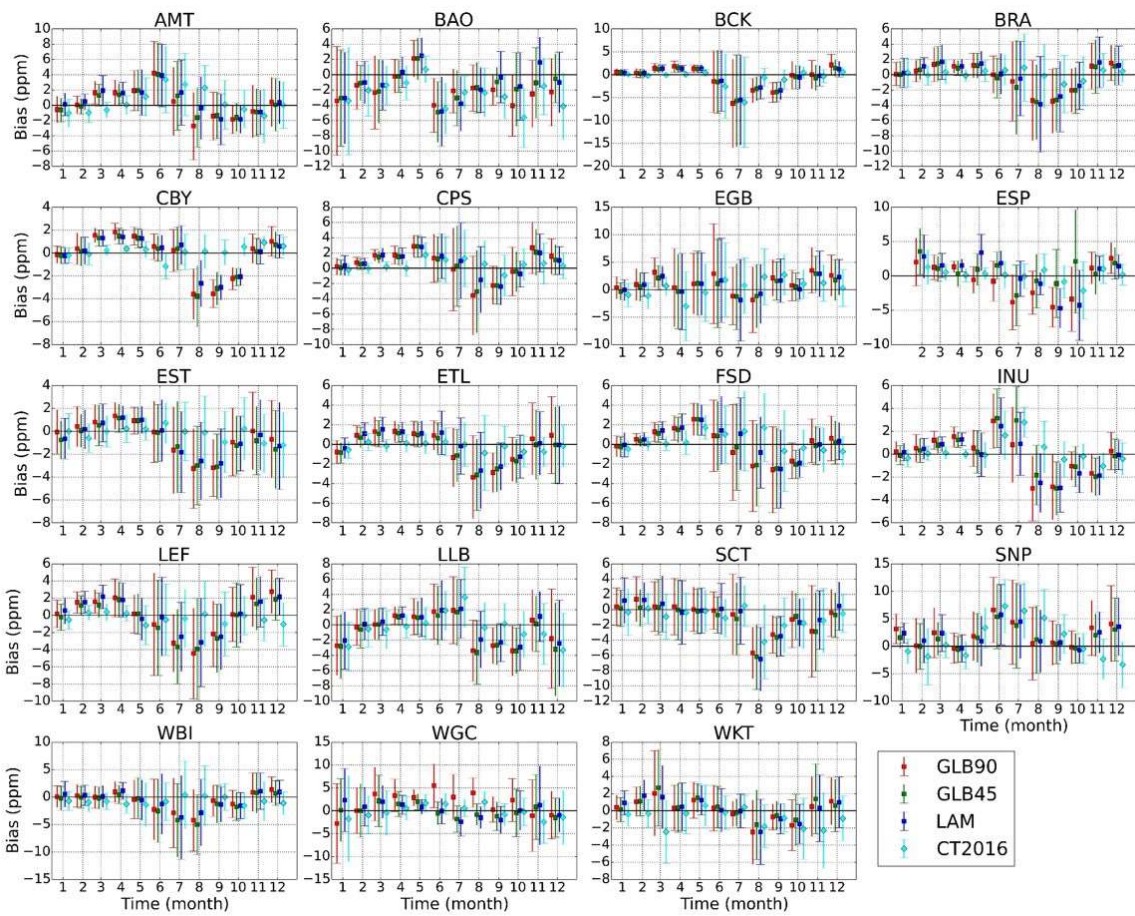

**Figure 7: Monthly mean bias of daily afternoon averaged (12-16 LST) CO₂ concentrations from GLB90 (red), GLB45 (green), LAM (blue) experiments and CT2016 (cyan) at all measurement sites used in this study.**

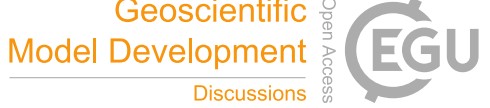



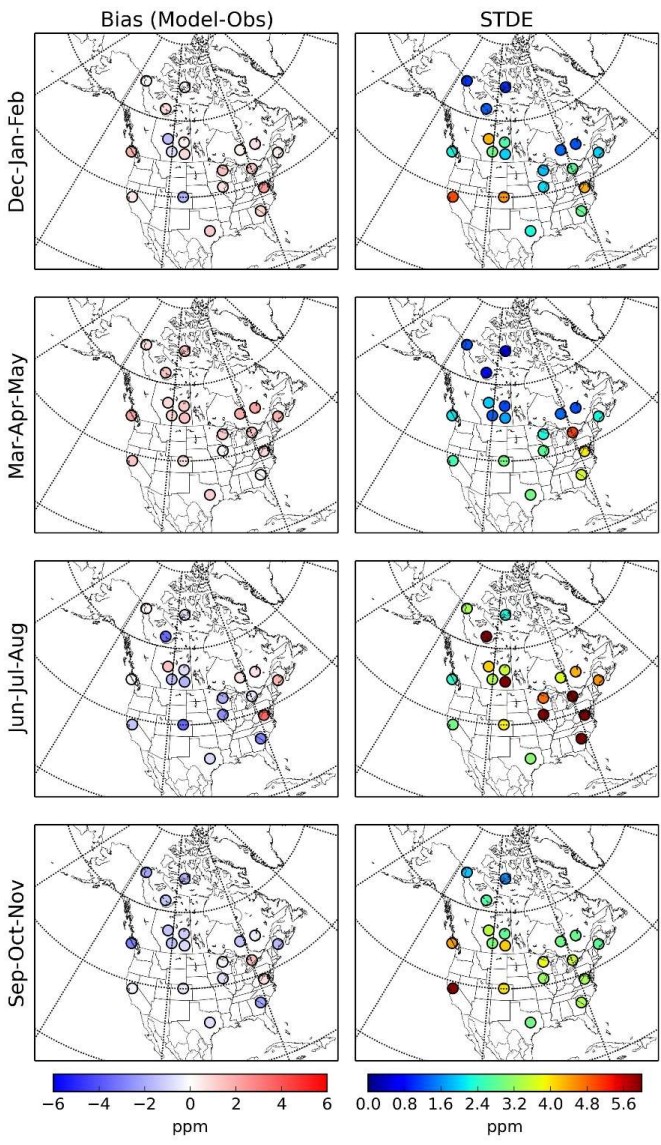

**Figure 8: Mean (left column) and standard deviation (right column) of the residuals between modelled CO$_2$ from LAM experiment and observed CO$_2$ concentrations (modelled – observed) at each observations site over January to February and December 2015 (first row), March to May 2015 (second row), June to August 2015 (third row) and September to November 2015 (fourth row). Should replace STD by STDE**





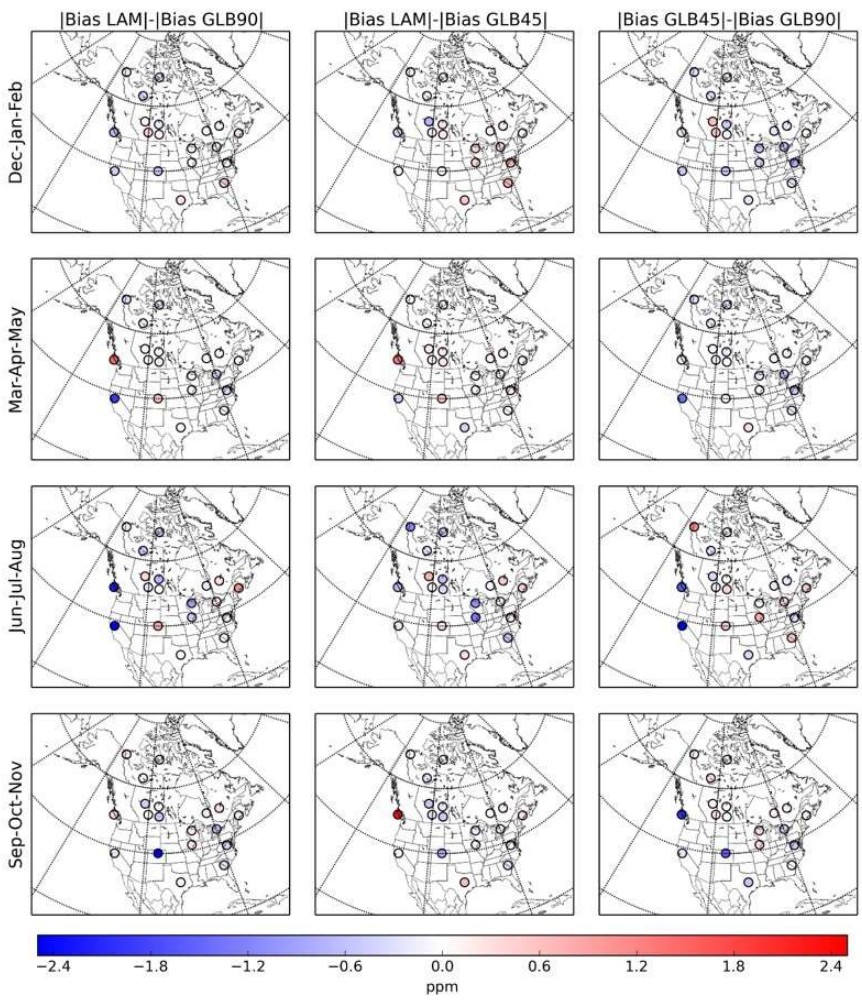

**Figure 9: The difference of absolute mean bias between GLB90 and LAM (first column), GLB45 and LAM (second column) and GLB90 and GLB45 (third column) over January to February and December 2015 (first row), March to May 2015 (second row), June to August 2015 (third row) and September to November 2015 (fourth row).**



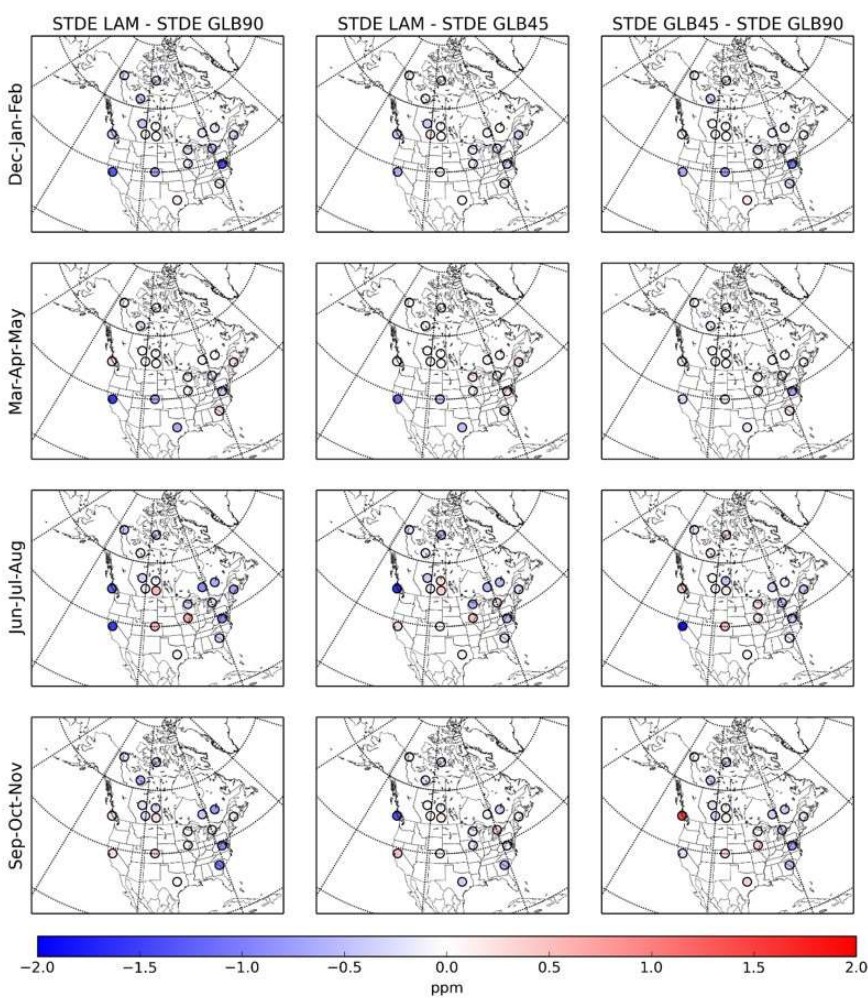

**Figure 10: The difference of standard deviation of the residuals between modelled CO$_2$ and observed CO$_2$ concentrations (modelled – observed) between GLB90 and LAM (first column), GLB45 and LAM (second column) and GLB90 and GLB45 (third column) over January to February and December 2015 (first row), March to May 2015 (second row), June to August 2015 (third row) and September to November 2015 (fourth row).**





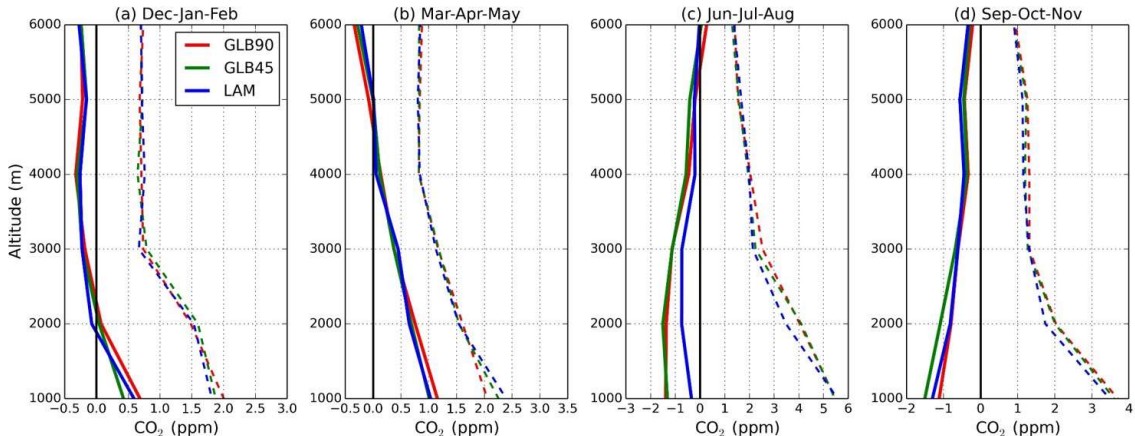

**Figure 11: Comparison of profiles of modelled CO₂ concentrations from GLB90 (red), GLB45 (green) and LAM (blue) experiments to NOAA aircraft observations for (a) January to February and December 2015, (b) March to May 2015, (c) June to August 2015 and (d) September to November 2015. Solid line denotes mean bias and dashed line denotes standard error. Sites used are Briggs**

5    **dale, Colorado; Cape May, New Jersey; Dahlen, North Dakota; Estevan Point, British Columbia; East Trout Lake, Saskatchewan; Homer, Illinois; Park Falls, Wisconsin; Worcester, Massachusetts; Poker Flat, Alaska; Charleston, South Carolina; Southern Great Plains, Oklahoma; Sinton, Texas; Trinidad Head, California; West Branch, Iowa.**



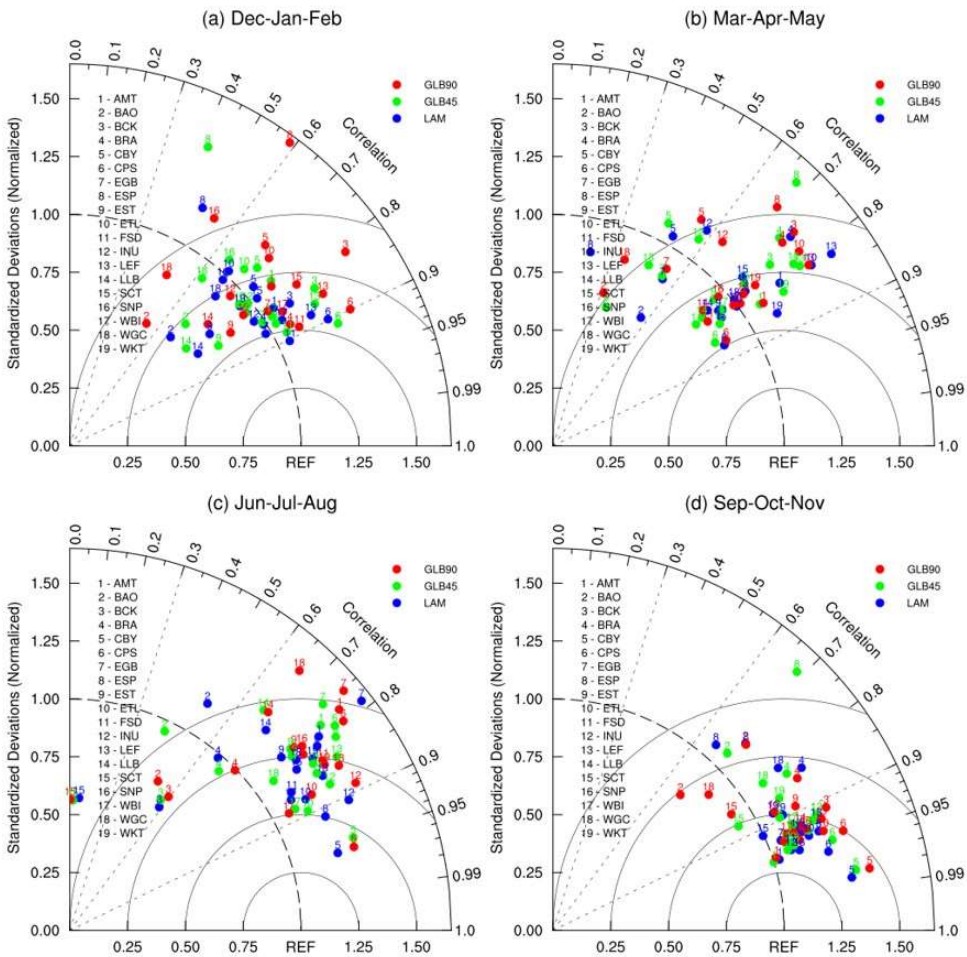

**Figure 12: Taylor diagram showing correlations and normalised standard deviations between daily afternoon modelled CO₂ concentrations from GLB90 (red) and GLB45 (green) and LAM (blue) experiments and observed CO₂ concentrations over (a) January to February and December 2015, (b) March to May 2015, (c) June to August 2015 and (d) September to November 2015.**



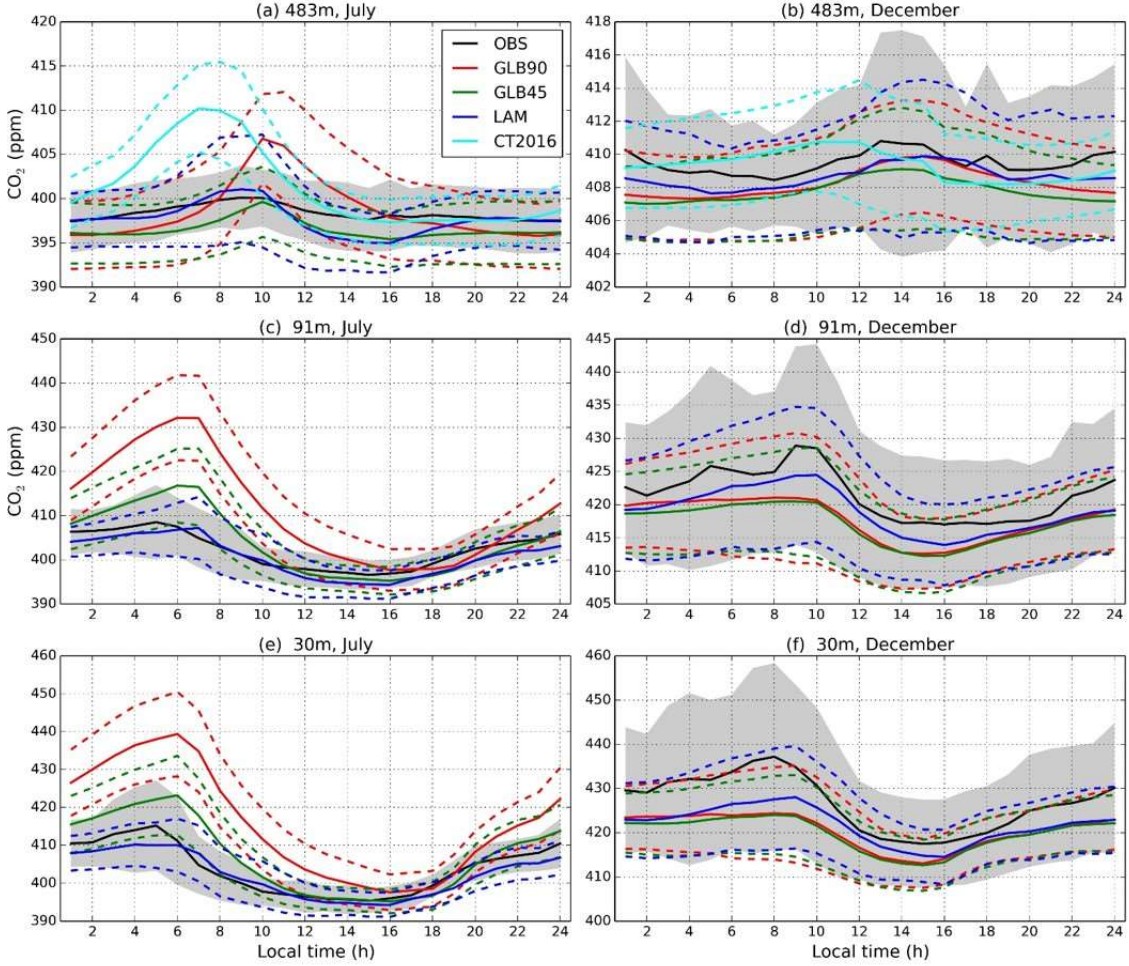

**Figure 13: Mean diurnal cycle of observed CO₂ concentrations (black) and modelled CO₂ concentrations from the GLB90 (red), GLB45 (green), LAM (blue) experiments and CT2016 (cyan) at WGC (Walnut Grove, California) for the intake height at (a) 483 m for July 2015 and (b) 483 m for December 2015, (c) 91 m for July 2015 (d) 91 m for December 2015, (e) 30 m for July 2015 and (f) 30 m for December 2015. The grey shaded region indicates 1 standard deviation above and below observed CO₂ concentrations while the dashed lines indicate the same for modelled CO₂ concentrations. Note that CT2016 results are only available at 483 m.**

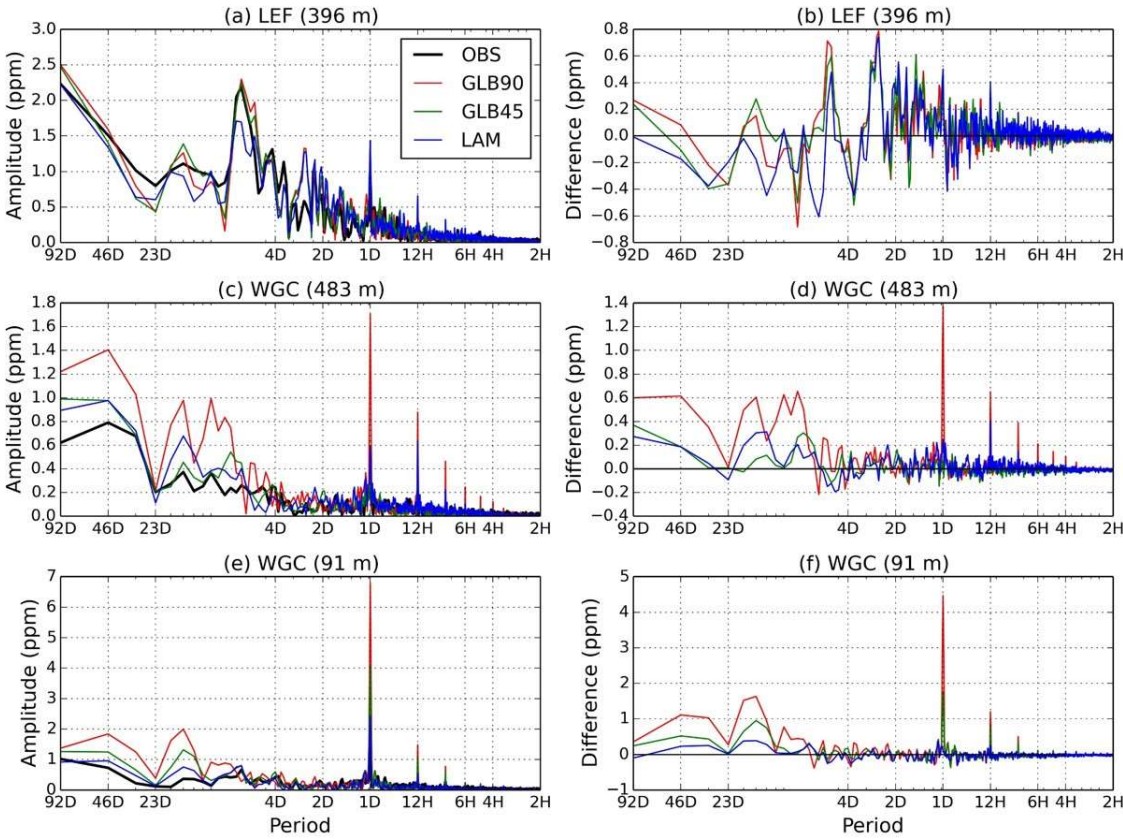

**Figure 14: The amplitude of hourly time series of observed CO₂ (black) and modelled CO₂ concentrations from GLB90 (red), GLB45 (green) and LAM (blue) experiments across temporal scales from 2 h to 92 days at (a) LEF (the intake height at 396 m) and (b) their differences, (c) WGC (intake height at 483 m) and (d) their difference and (e) WGC (intake height at 91 m) and (f) their differences.**



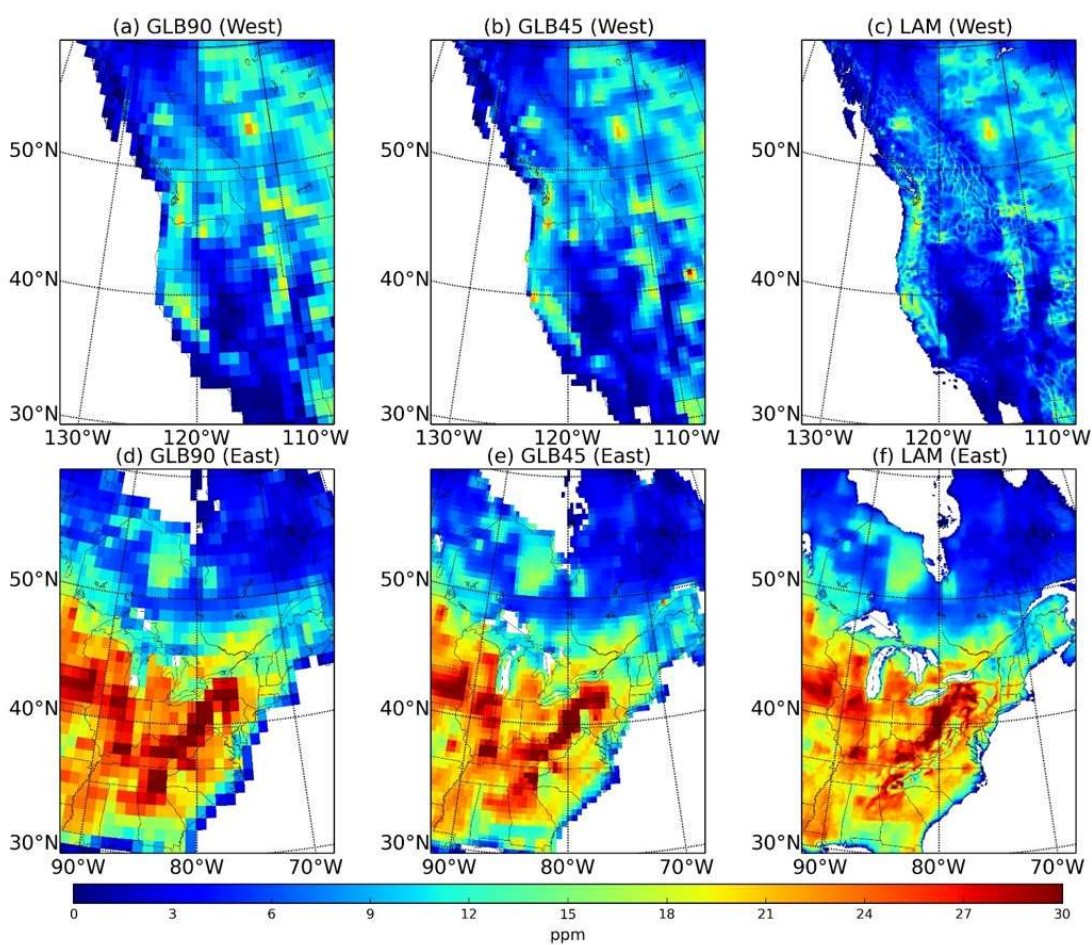

**Figure 15: The amplitude of the diurnal cycle of hourly time series of $CO_2$ concentrations at the lowest model level from the (a,d) GLB90, (b,e) GLB45 and (c,f) LAM experiments during June to August 2015. Panels show zoomed regions over the west part of North America (a-c) and the east part of North America (d-f).**