# Peer review of "The Canadian atmospheric transport model for simulating greenhouse gas evolution on regional scales: GEM-MACH-GHG v.137-reg"

_Geoscientific Model Development, 2019_

## Referee Comment (RC1) · Anonymous Referee #1 · 14 Aug 2019

Review of Kim et al.: The Canadian atmospheric transport model for simulating greenhouse gas evolution on regional scales: GEM-MACH-GHG v.137-reg

General comments: the authors have developed an integrated greenhouse gas transport system from the global to regional scales. In this study, the regional domain of interest they chose is Canada and the United States. Three experiments were set up and evaluated with various meteorological and CO2 data. These experiments are GLB90, GLB45, LAM. GLB90 is the global model output with 0.9 deg grid spacing, GLB45 is similar to GLB90 but with 0.45 deg grid spacing, and LAM is the regional

setup at the 10 km resolution driven with GLB45' meteorology and CO2. All of the experiments used the same surface fluxes from CT2016. That the main message of this study delivers is the high-resolution outperforms the simulations.

The content of the manuscript lies in the category of the model description papers of GMD and meets the requirement of this category. The manuscript is generally well written, and the analyses presented are fairly well, I would like to recommend for publication after addressing my concerns as follows.

1. The CT2016 posterior CO2 mixing ratios are used as a baseline in Figure 7. It seems to me that the transport difference causes much bigger discrepancy than those by grid spacing. Both of Carbon Tracker and GEM-MACH-GHG are an operational system, and this is a forefront work towards (regional) flux estimates. One recent study on the transport uncertainty (Schuh et al, 2019) demonstrates the large difference in CO2 mixing ratios between TM5 and Geos-Chem models using the same CT surface fluxes. They show that the inverse model flux estimates for large zonal bands can have systematic biases of up to 1.7 PgC/year due to large-scale transport uncertainty. The difference between CT2016 and GEM-MACH-GHG models in Figure 7 implies the bias in the GEM-MACH-GHG transport. I would expect the authors to have a discussion on the possible strategies to improve GEM-MACH-GHG transport since it is an operational system as Carbon Tracker. Additionally, my personal experience of working with CT2016 and CT2017 is that CT2017 outperforms the posterior CO2 mixing ratios a lot over CT2016. I encourage the authors to add CT2017 results in Figure 7 as well.

2. The authors use site codes quite often in writing. It may cause difficulties and confusion to some readers who are not familiar with the geographic locations of those sites to follow the statements. I recommend the authors to add the codes of the stations in Figure 1 for the references.

3. In the manuscript, the authors gave the credit to LAM for resolving the complex
terrain better. This is an overstatement. I don't see clear improvement at the mountain sites (such as BAO, in Figure 9 and 10) or coastal sites (such as ESP, in Figure 9 and 10). I was puzzled by that because I also expected the outperformance from the high-resolution simulations. One of the reasons that the improvement of the high-resolution simulations is not the evidence is that the analyses/figures shown were averaged over a relatively long period (monthly or seasonally). The improvement should be more evident in a higher frequent timescale, such as daily or even hourly due to resolving faster physics. I recommend the authors to select a or a few cases and to show hourly/daily time series for that/those case(s), as Figure 15 in Agustí-Panareda, et al., 2019, by which the authors should be able to justify the statement better.

Specific comments:

1. P2/L12, add a more recent work from OCO2MIP (Crowell et al., 2019)

2. P16/L20, "from Figure 15, it can be concluded that higher horizontal resolution might help to enhance the performance of CO2 simulations even without using fluxes on a finer grid spacing " is an overstatement. I don't see the statement is true. Please clarify it. Even though the observation network is sparse, there are still a few sites in the domain of interest. I would like to see the same matrices from the observations overlaid to the model values in Figure 15.

3. P17/L28, "This is a promising result because it suggests that using night-time data in an inversion to estimate night time fluxes (e.g. Lauvaux et al., 2008) may be beneficial if a high-resolution model is used. " This can be demonstrated by using CT2017. CT2017 has assimilated nighttime data, but CT2016 doesn't.

4. P18/L33, add Feng et al, 2016, another study demonstrates that both of the high-resolution transport and fluxes are demanding for accurate CO2 simulations at the urban scale. It also links to the last paragraph of the conclusion.

5. P19/L7, add Díaz-Isaac et al., 2018, which demonstrate that the meteorological
IC/LBC is one of the big players in regional CO2 simulations.

6. Figure 1, add the codes of the observation sites.

7. Figure 7, it's a very dense figure. The panels should be enlarged. To make the results stand out, the authors should use lines instead of markers for presentation.

8. Figure 8, remove "Should replace STD by STDE" in the caption.

9. Figure 12, replace the greens with another color. The green numbers are not distinguishable in most of the cases. The figures should be enlarged.

10. Figure 14, can the author extend the time scale from 92 D to at least half year? It should be very interesting results from hourly to a half year time scale.

11. Figure 15, add "in" following "zoomed" in the caption.

Reference: Agustí-Panareda, A., Diamantakis, M., Massart, S., Chevallier, F., Muñoz-Sabater, J., Barré, J., Curcoll, R., Engelen, R., Langerock, B., Law, R. M., Loh, Z., Morguí, J. A., Parrington, M., Peuch, V.-H., Ramonet, M., Roehl, C., Vermeulen, A. T., Warneke, T., and Wunch, D.: Modelling CO2 weather - why horizontal resolution matters, Atmos. Chem. Phys., 19, 7347-7376, https://doi.org/10.5194/acp-19-7347-2019, 2019. Crowell, S., Baker, D., Schuh, A., Basu, S., Jacobson, A. R., Chevallier, F., Liu, J., Deng, F., Feng, L., McKain, K., Chatterjee, A., Miller, J. B., Stephens, B. B., Eldering, A., Crisp, D., Schimel, D., Nassar, R., O'Dell, C. W., Oda, T., Sweeney, C., Palmer, P. I., and Jones, D. B. A.: The 2015-2016 carbon cycle as seen from OCO-2 and the global in situ network, Atmos. Chem. Phys., 19, 9797-9831, https://doi.org/10.5194/acp-19-9797-2019, 2019. Díaz-Isaac, L. I., Lauvaux, T., and Davis, K. J.: Impact of physical parameterizations and initial conditions on simulated atmospheric transport and CO2 mole fractions in the US Midwest, Atmos. Chem. Phys., 18, 14813-14835, https://doi.org/10.5194/acp-18-14813-2018, 2018. Feng, S., Lauvaux, T., Newman, S., Rao, P., Ahmadov, R., Deng, A., Díaz-Isaac, L. I., Duren, R. M., Fischer, M. L., Gerbig, C., Gurney, K. R., Huang, J., Jeong, S., Li, Z., Miller, C. E., O'Keeffe, D., Patarasuk, R.,

GMDD
Sander, S. P., Song, Y., Wong, K. W., and Yung, Y. L.: Los Angeles megacity: a highresolution land-atmosphere modelling system for urban CO2 emissions, Atmos. Chem. Phys., 16, 9019-9045, https://doi.org/10.5194/acp-16-9019-2016, 2016. Schuh, A. E., Jacobson, A. R., Basu, S., Weir, B., Baker, D., Bowman, K., et al. (2019). Quantifying the impact of atmospheric transport uncertainty on CO2 surface flux estimates. Global Biogeochemical Cycles, 33, 484-500. https://doi.org/10.1029/2018GB006086

---

## Referee Comment (RC2) · Anonymous Referee #2 · 22 Oct 2019

This manuscript describes the set-up of a new Limited Area Model for the modelling of atmospheric CO2 concentrations as part of the wider range of models at ECCC. The LAM is compared to the global model to assess the benefit of modelling at higher spatial resolution over the North American domain. The paper is mostly descriptive, although the Discussion draws some conclusions and presents some recommendations. As such, it does fit the scope of GMD. The manuscript fits also very well in the increasing debate about transport errors in the (inverse) modelling of CO2. Various groups are pushing for increased resolution to minimise the transport errors and thre

is also a discussion about the advantage of LAM models versus of-line models. The manuscript is well-written and everything is clearly explained.

The authors have put together a LAM with similar characteristics as their global model, which is indeed a very sensible way to go. In my view, the paper would have been more interesting, if a regional off-line model would have been included in the study as well. This would have allowed for the additional assessment of using a LAM versus using an off-line model in terms of transport errors through for instance interpolation approximations. I do realise, however, that this would have significantly increased the work, so maybe it can be considered for future work. For this paper, it would be good to have an additional paragraph on this consideration.

My main issue is with the quality of the figures. Some of them are difficult to read (e.g., Figure 7), have washed-out colours (e.g., Figures 8 - 10), or contain too many curves or points (e.g., Figures 12 and 13). I encourage the authors to look at these figures again to see if they can be simplified or made clearer.

Some minor comments: Section 4.1, first paragraph: while I can fully understand why the STDE would be reduced with increased resolution, it is not directly obvious why the bias should be different between the different resolutions. It would be useful to comment a bit more on this in this paragraph. Page 15, line 20: this should be Section 2.5 Page 19, lines 1 - 5: it would be good to refer here to the 2019 Agusti paper as well, because it deals with the flux resolution problem by explicitly modelling these fluxes at the same resolution as the transport model.
* * *

---

## Author Comment (AC1) · 21 Nov 2019

**Response to Reviewer 1**

Reviewer's comment is in black. Our responses are in blue text. The references are to the manuscript with changes tracked.

General comments: the authors have developed an integrated greenhouse gas transport system from the global to regional scales. In this study, the regional domain of interest they chose is Canada and the United States. Three experiments were set up and evaluated with various meteorological and CO2 data. These experiments are GLB90, GLB45, LAM. GLB90 is the global model output with 0.9 deg grid spacing, GLB45 is similar to GLB90 but with 0.45 deg grid spacing, and LAM is the regional setup at the 10 km resolution driven with GLB45' meteorology and CO2. All of the experiments used the same surface fluxes from CT2016. That the main message of this study delivers is the high-resolution outperforms the simulations. The content of the manuscript lies in the category of the model description papers of GMD and meets the requirement of this category. The manuscript is generally well written, and the analyses presented are fairly well, I would like to recommend for publication after addressing my concerns as follows.

**Response**: We appreciate the reviewer's careful reading of our manuscript and helpful comments. We have revised the manuscript following the reviewer's suggestions.

1. The CT2016 posterior CO2 mixing ratios are used as a baseline in Figure 7. It seems to me that the transport difference causes much bigger discrepancy than those by grid spacing. Both of Carbon Tracker and GEM-MACH-GHG are an operational system, and this is a forefront work towards (regional) flux estimates. One recent study on the transport uncertainty (Schuh et al, 2019) demonstrates the large difference in CO2 mixing ratios between TM5 and Geos-Chem models using the same CT surface fluxes. They show that the inverse model flux estimates for large zonal bands can have systematic biases of up to 1.7 PgC/year due to large scale transport uncertainty. The difference between CT2016 and GEM-MACH-GHG models in Figure 7 implies the bias in the GEM-MACH-GHG transport. I would expect the authors to have a discussion on the possible strategies to improve GEM-MACH-GHG transport since it is an operational system as Carbon Tracker. Additionally, my personal experience of working with CT2016 and CT2017 is that CT2017 outperforms the posterior CO2 mixing ratios a lot over CT2016. I encourage the authors to add CT2017 results in Figure 7 as well.

**Response**:**

There are actually three separate points being made by the Reviewer in this first comment. We will address all 3 points in the next 3 paragraphs.

One point concerns the mismatch of GEM-MACH-GHG and TM5 transport errors. While the Reviewer states that the difference between 2 models "implies a bias in the GEM-MACH-GHG transport", in fact what we can infer is only a mismatch in transport between the two models. The reason for the mismatch was already discussed in Section. Page 17, lines 26-29: "As discussed in Polavarapu et al. (2016), GEM has different transport behaviour from the transport model used in CarbonTracker, in particular over the Arctic region, as seen in time series of CO2 concentrations and column-averaged CO2. Thus, our models are not expected to perform better than CT2016 because we use surface CO2 fluxes inferred by an inversion framework using a different transport model which has different transport behaviour". All experiments

(GLB90, GLB45 and LAM) used posterior  $CO_2$  fluxes from CT2016 that are inherently consistent with the transport behaviour of TM5 which is the transport model used in CT2016. Understandably, posterior  $CO_2$  mixing ratios in CT2016 take advantage of this consistency, producing better results (smaller biases in general) than the GEM-MACH-GHG. This baseline result is included just to show that our model behaviour does not diverge too much from the CT2016 result despite this handicap. Of course, the main purpose of this study is to report the performance of the developed regional version of GEM-MACH-GHG compared to its (global) variations. Our strategy to properly assess the transport behaviour of the GEM-MACH-GHG in estimated fluxes is to develop a flux estimation system based on the GEM-MACH-GHG as mentioned in the Section 5 (Page 18, lines 2-4). Only then will our model simulations with posterior fluxes be consistent with the model dynamics used to generate the posterior fluxes, and the comparison to observations fairer. This development is currently in progress.

Another point concerns the use of CT2017 instead of CT2016 in Figure 7. The correct comparison for our Figure 7 is CT2016 because our model actually used CT2016 fluxes to produce the mixing ratios. Nevertheless, following the reviewer's suggestion, we looked at CT2017 results in addition to CT2016 results in the same style as Figure 7 in the manuscript (Figure R1), excluding the GEM-MACH-GHG results to focus on the comparison between two CT versions. It is difficult to say that CT2017 outperforms than CT2016 with respect to the posterior  $CO_2$  mixing ratio, in terms of monthly bias and its standard deviation, at least for the afternoon (12-16 LST) data in 2015. On the other hand, looking at the results for every hour data (Figure R2), CT2017 outperforms CT2016 in terms of  $CO_2$  bias, as the Reviewer expected. However, since we used fluxes from CT2016 which assimilated afternoon data in its inversion, the appropriate comparison is to CT2016 and afternoon time data in Figure 7. Since it would further crowd Figure 7 and distract from the main point of the figure, we decided not to add CT2017 results in figure 7. Instead, we have revised the manuscript to better clarify the motivation of using CT2016.

**Page 11, lines 20-24**: The CO2 fields in the LAM and other experiments are investigated in terms of monthly bias and STDE of daily afternoon (12:00-16:00 LST) modelled CO2 concentrations at the measurement sites shown in Fig. 1 and listed in Table 1 (Fig. 7). Daily afternoon time was selected because this is what CT2016 used to estimate surface CO2 fluxes. Also, CT2016 results are included as a reference in order to verify results of three experiment simultaneouslysince this is what all our model experiments used as input fluxes.

A final point concerns the fact that transport error may be large compared to error of model resolution. On this point, we concur. In the future, our flux estimation system development will solve the problem of the mismatch of transport error with TM5. However, we will still have the problem of transport error since every transport model has this issue. In the future, we plan to address this issue by using different sources of meteorology in the flux estimation system. This point is now mentioned in the discussion section.

**Page 18, lines 3-7**: When posterior fluxes become available from our global model, this will alleviate the issue of model transport error mismatches with CarbonTrackerbeing embedded in the surface fluxes. However, we will still have transport error, which is one of the biggest sources of posterior uncertainties in an inversion (Schuh et al., 2019). To address this issue, we plan to use multiple sources of meteorology to better account for transport error in posterior flux and uncertainty estimates.

Schuh, A. E., Jacobson, A. R., Basu, S., Weir, B., Baker, D., Bowman, K., Chevallier, F., Crowell, S., Davis, K. J., Deng, F., Denning, S., Feng, L., Jones, D., Liu, J., and Palmer, P. I.: Quantifying the Impact of Atmospheric Transport Uncertainty on CO2 Surface Flux Estimates, Global Biogeochem. Cy., 33, 2018GB006086, https://doi.org/10.1029/2018GB006086, 2019.

---

## Author Comment (AC2)

**Response to Reviewer 2**

Original text is in black. Our responses are in blue text. The references are to the manuscript with changes tracked.

This manuscript describes the set-up of a new Limited Area Model for the modelling of atmospheric CO2 concentrations as part of the wider range of models at ECCC. The LAM is compared to the global model to assess the benefit of modelling at higher spatial resolution over the North American domain. The paper is mostly descriptive, although the Discussion draws some conclusions and presents some recommendations. As such, it does fit the scope of GMD. The manuscript fits also very well in the increasing debate about transport errors in the (inverse) modelling of CO2. Various groups are pushing for increased resolution to minimise the transport errors and there is also a discussion about the advantage of LAM models versus of-line models. The manuscript is well-written and everything is clearly explained.

**Response**: We appreciate reviewer's careful reading of our manuscript and helpful comments. We have revised the manuscript following the reviewer's suggestions.

The authors have put together a LAM with similar characteristics as their global model, which is indeed a very sensible way to go. In my view, the paper would have been more interesting, if a regional off-line model would have been included in the study as well. This would have allowed for the additional assessment of using a LAM versus using an off-line model in terms of transport errors through for instance interpolation approximations. I do realise, however, that this would have significantly increased the work, so maybe it can be considered for future work. For this paper, it would be good to have an additional paragraph on this consideration.

**Response**: Following the reviewer's suggestion, we have added an additional paragraph in the section 5 (Discussion and conclusions).

**Page 19, lines 9-14**: While this work has focused on the benefit of our higher resolution regional model over our global model for $CO_2$ simulation, both models are "online" in that the meteorology is coupled to the tracer transport every time step. An interesting question that was not addressed here is the impact of increased horizontal resolution in the context of an "offline" transport model which ingests meteorological analyses or reanalyses from another model (e.g. Kjellström et al., 2002; Geels et al., 2004, 2007). Additional errors arise due to spatial and temporal interpolation from another model's grid to the offline model's grid then arise.

Geels, C., Doney, S., Dargaville, R., Brandt, J., and Christensen, J.: Investigating the sources of synoptic variability in atmospheric $CO_2$ measurements over the Northern Hemisphere continents: a regional model study, Tellus B, 56, 35–50, 2004.
Geels, C., Gloor, M., Ciais, P., Bousquet, P., Peylin, P., Vermeulen, A. T., Dargaville, R., Aalto, T., Brandt, J., Christensen, J. H., Frohn, L. M., Haszpra, L., Karstens, U., R¨oödenbeck, C., Ramonet, M., Carboni, G., and Santaguida, R.: Comparing atmospheric transport models for future regional inversions over Europe – Part 1: Mapping the atmospheric $CO_2$ signals, Atmos. Chem. Phys., 7, 3461–3479, doi:10.5194/acp-7-3461-2007, 2007.

Kjellström, E., Holmén, K., Eneroth, K., and Engardt, M.: Summertime Siberian $CO_2$ simulations with the regional transport model MATCH: a feasibility study of carbon uptake calculations from EUROSIB data, Tellus, 54B, 834-849, https://doi.org/10.3402/tellusb.v54i5.16733, 2002.

My main issue is with the quality of the figures. Some of them are difficult to read (e.g., Figure 7), have washed-out colours (e.g., Figures 8 - 10), or contain too many curves or points (e.g., Figures 12 and 13). I encourage the authors to look at these figures again to see if they can be simplified or made clearer.

**Response**: Following the reviewer's suggestions (reviewer #1 also had similar concerns), we looked at figures again and have improved the quality of figures (Fig. 7 – 10, 12 and 13).
- Figure 7: The figure was enlarged and lines were made thicker.
- Figures 8, 9 and 10: The range of colours (including label bar) was narrowed down in order to vivify colours. Of course, some dots are still transparent since their values are close to zero, i.e. small differences between two results.
- Figure 12: This figure was enlarged and the light green numbers was replaced by darker green to make the green numbers more visible.
- Figure 13: Some lines were replaced by shading to make them clearer.

Some minor comments:
Section 4.1, first paragraph: while I can fully understand why the STDE would be reduced with increased resolution, it is not directly obvious why the bias should be different between the different resolutions. It would be useful to comment a bit more on this in this paragraph.

**Response**: To clarify why the bias is reduced. We have revised the first paragraph in Section 4.1

**Page 13, lines 3-5**: In JJA, the reduction in bias resulting from the higher horizontal resolution model can be seen clearly and the magnitude of reduction is higher probably due to better weather simulation (less transport errors) as shown in Figs. 3, 4, 5 and 6.

Page 15, line 20: this should be Section 2.5

**Response**: Yes. We have correct the typo.

Page19, lines1-5: it would be good to refer here to the 2019 Agusti paper as well, because it deals with the flux resolution problem by explicitly modelling these fluxes at the same resolution as the transport model.

**Response**: Following the reviewer's suggestion, we have added a statement, referring to Agustí-Panareda et al. (2019) in the proper place.

**Pager 19, lines 20-22**: One way to deal with this issue is to model biogenic fluxes explicitly at the same horizontal resolution as the transport model (e.g. Agustí-Panareda et al., 2019). Indeed, this is an avenue we plan to investigate in the future.

Agustí-Panareda, A., Diamantakis, M., Massart, S., Chevallier, F., Muñoz-Sabater, J., Barré, J., Curcoll, R., Engelen, R., Langerock, B., Law, R. M., Loh, Z., Morguí, J. A., Parrington, M., Peuch, V.-H., Ramonet, M., Roehl, C., Vermeulen, A. T., Warneke, T., and Wunch, D.: Modelling $CO_2$ weather – why horizontal resolution matters, Atmos. Chem. Phys., 19, 7347–7376, https://doi.org/10.5194/acp-19-7347-2019, 2019.